# A Robust Differential Neural ODE Optimizer

**Panagiotis Theodoropoulos   Guan-Horng Liu   Tianrong Chen   Augustinos D. Saravanos**
**Evangelos A. Theodorou**
Georgia Institute of Technology, USA
*{ptheodor3, ghliu, tianrong.chen, asaravanos3, evangelos.theodorou}@gatech.edu*

## Abstract

Neural networks and neural ODEs tend to be vulnerable to adversarial attacks, rendering robust optimizers critical to curb the success of such attacks. In this regard, the key insight of this work is to interpret Neural ODE optimization as a min-max optimal control problem. More particularly, we present Game Theoretic Second-Order Neural Optimizer (GTSONO), a robust game theoretic optimizer based on the principles of min-max Differential Dynamic Programming. The proposed method exhibits significant computational benefits due to efficient matrix decompositions and provides convergence guarantees to local saddle points. Empirically, the robustness of the proposed optimizer is demonstrated through greater robust accuracy compared to benchmark optimizers when trained on clean images. Additionally, its ability to provide a performance increase when adapted to an already existing adversarial defense technique is also illustrated. Finally, the superiority of the proposed update law over its gradient based counterpart highlights the potential benefits of incorporating robust optimal control paradigms into adversarial training methods.

## 1 Introduction

Despite the remarkable performance achieved by deep learning based models on diverse challenges from image classification to speech recognition, recent findings indicate that deep neural networks tend to be fragile to adversarial examples(Liu et al. (2020)). This problem of generating adversarial examples and conversely training models that are not easily misguided by these samples has gathered significant attention (Carrara et al. (2019)). However, the development of robust models has been proven to be more challenging (Tsipras et al. (2018)), rendering adversarial robustness a central object of study in machine learning (Carlini et al. (2019)). This study has led to significant advance in understanding the generalization and interpretability behind the learning capacity of these models (Huang et al. (2022)). Traditionally, the most widely used approach to building adversarially robust models has been adversarial training in which the classifier is trained on both adversarial and clean samples aiming to generalize well in both instances(Madry et al. (2017); Wong & Kolter (2018); Raghunathan et al. (2018); Goodfellow et al. (2014)). The goal of adversarial training is to find the saddle point that solves the following optimization problem

$$\min_{\theta} \mathbb{E}\left[ \max_{\delta \in S} \mathcal{L}(\mathbf{x} + \delta, \mathbf{y}; \theta) \right] \tag{1}$$

where $\mathbf{x} \in \mathbb{R}^m$ is the state vector and $\mathbf{u} \in \mathbb{R}^n$ is the control, the perturbation term $\delta$ is bounded by $\sup ||\delta||^2 \leq \beta^2$ and belongs to set $S$, which is the set of admissible perturbations, and $\mathcal{L}(\mathbf{x} + \delta, \theta)$ is the definition of the loss function. Although adversarial training is effective, there are some drawbacks such as the considerable increase in the training time.

In the work by Poznyak et al. (2023), it was suggested that a min-max robust control can be viewed as a neural dynamic programming approach using continuous differential neural networks. In this work, the peturbations $\delta$ were recasted, yielding the same Hamilton Jacobi Bellman equation as in the case of the game theoretic Differential Dynamic Programming (DDP) (Jacobson & Mayne (1970); Sun et al. (2018)).

In the realm of Optimal Control (OC), it has been demonstrated that game-theoretic formulation enables the model handling uncertainties and external disturbances Sun et al. (2015). This algorithm has been successfully tested in real-life robotic applications, coming up up with robust control policies

for robotic systems operating under unknown external disturbances Sun et al. (2018), Morimoto et al. (2003). This paves the way to draw connections between adversarial defense methods and robust optimal control and design principled algorithms that demonstrate the capacity to handle disturbances.

In this work, drawing inspiration by the connection of training Neural Networks and continuous time OCP Liu et al. (2021a;b), we introduce a novel second order Game Theoretic Neural Optimizer, by recasting the training of Neural ODEs as trajectory optimization problem through this Min-Max OCP paradigm. Our approach differs considerably from conventional min-max methods for adversarial learning. While most min-max adversarial training methods consider the adversary input, our proposed method instead consider the disturbance being injected through the antagonizing control. The resulting framework, called GTSONO, is designed based on the continuous time Min-Max DDP algorithm engendering a innately robust Neural Optimizer. This optimizer is capable of handling uncertainties and finding a control policy or equivalently a weight configuration that robustifies the network and enables it to resist adversarial attacks. Our analysis provides theoretic guarantees that our algorithm converges to a saddle point, and our experiments empirically verify that our algorithm converges to a saddle point even when trained with clean images, which is translated to an increased robustness of our model against adversarial attacks, even with natural training. Our contribution can be summarized in the following points:

- We propose a novel game theoretic optimizer for training robust neural ODEs based on the principles of min-max , reformulating the input adversary as the antagonizing control to model system disturbances. An efficient computational framework is presented based on decomposing matrices into vectors, which can be easily obtain through solving the ODE.

- We illustrate that the proposed optimizer is guaranteed to converge to stable local saddle points under mild assumptions.

- We empirically showcase that GTSONO increases the robustness and generates more confident predictions compared to benchmark optimizers on natural training.

- We demonstrate that GTSONO is successfully adapted to already existing adversarial training methods, enhancing their robustness. The second order convergence nature of min-max DDP enables our optimizer to converge in fewer epochs, and lower wall-clock time, which highlights its computational efficiency.

## 2  RELATED WORKS

In order to view the landscape of the pertinent literature, we review the advancements in both fronts.

**Adversarial Attacks.** A number of adversarial attack methods have been developed, such as optimization based (Szegedy et al. (2013)), gradient-based (Moosavi-Dezfooli et al., 2016; Xie et al., 2019; Dong et al., 2018), and attacks from Generative Networks (Baluja & Fischer, 2017; Zhao et al., 2017). Traditionally, the most famous gradient based methods are the Fast Gradient Sign Method (FGSM) (Goodfellow et al., 2014) and the Projected Gradient Descent (PGD) (Madry et al., 2017). The $\ell_\infty$ PGD attack is used to undermine the estimations and predictions of machine learning models, by iteratively perturbing input data in the direction of the gradient, while constraining the perturbations to stay within a predefined projection set determined by parameter $\epsilon$. Conversely, FGSM is a more efficient gradient based attack, which operates by taking a single step in the direction of the gradient of the loss function with respect to the input. Therefore, the attack efficiency of FGSM is higher at the expense of its success rate, compared to the iterative PGD. Compared with other algorithms, network-generated adversarial examples have been observed to be deceptive, however their efficiency is deemed low, since they require multiple rounds of iterations for the optimal solution (Liang et al., 2022). Optimization based attacks i.e. Carlini & Wagner (CW; Carlini & Wagner (2017)) have been found to be more effective to certain defensive schemes, such as defensive distillation (Liang et al., 2022) and masking (Huang et al., 2022).

**Adversarial Defense.** In order to deal with the threat of adversarial attacks, on the one hand, researchers try to improve the model's resistance to these attacks, developing numerous defense methods, so that the model can make correct predictions on adversarial samples (Wang et al., 2022). Examples of such defense methods are defensive distillation (Papernot et al., 2016), gradient regularization (Papernot et al., 2017; Ross & Doshi-Velez, 2018), and adversarial training (Tramèr et al., 2017; Zhang et al., 2019a; Zheng et al., 2020), which is traditionally the most successful.

Most notably, this technique attempts to improve the robustness of a neural network by training it with adversarial samples, i.e., generated by FGSM or PGD. However, the issue with this defense mechanism is the computational overhead. To speed up and solve this problem, Adversarial Training for Free (FreeAT) (Shafahi et al., 2019) was proposed.

Recently more methods have been developed such as masking and denoising. In fact, many attack methods rely on the gradient information of the victim model. In gradient masking/obfuscation defense methods try to defend by hiding the gradient information (Naseer et al., 2019). Moreover, Huang et al. claimed that adaptive stepsize numerical ODE solver, such as DOPRI5, has a gradient masking effect that fails the PGD attacks which are dependent on gradient information. This implies that continuous neural models possess innate robustness compared to discrete neural networks. However, this method cannot fool gradient-free attacks such as CW and SPSA. On the other hand, since the generation of adversarial examples takes place by adding specific noise; image denoising is another technique to enable the models to resist adversarial attacks. Ordinary denoisers suffer from low performance due to error amplification effect. In this regard, Liao et al. proposed a high-level representation guided denoiser (HGD) suppressing the amplification of errors.

## 3 METHODOLOGY

### 3.1 TRAINING NEURAL ODES USING GAME THEORETIC OPTIMAL CONTROL THEORY

First, Neural ODEs (Chen et al. (2018)) generally concern the following optimization over an objective function: $\mathcal{L}$:

$$\min_\theta \mathcal{L}(\mathbf{x}(t_f)), \quad \text{where} \quad \frac{d\mathbf{x}}{dt} = F(t, \mathbf{x}(t), \theta), \quad \mathbf{x}(t_0) = \mathbf{x}_0 \tag{2}$$

where $\mathbf{x}(t) \in \mathbb{R}^m$, and $F(\cdot, \cdot, \theta)$ is a deep neural network parameterized by $\theta \in \mathbb{R}^n$. In the proposed methodology, Eq. (2) is recast as a trajectory optimization problem, through the perspective of Min-Max OC, by introducing a second set of antagonizing controls (e.g. weights in DNN) that attempt the maximization of the loss function. This second set of parameters aims to represent the existence of disturbances providing the robustness of our model. Finally, for this problem, we obtain:

$$\min_{\mathbf{u}} \max_{\mathbf{v}} \left[ \Phi(x_{t_f}) + \int_{t_0}^{t_f} \ell(t, \mathbf{x}, \mathbf{u}, \mathbf{v}) dt \right], \quad \text{s.t.} \begin{cases} \frac{d\mathbf{x}}{dt} = F(t, \mathbf{x}, \mathbf{u}, \mathbf{v}), & \mathbf{x}(t_0) = \mathbf{x}_0 \\ \frac{d\mathbf{u}}{dt} = 0, & \mathbf{u}(t_0) = \theta \\ \frac{d\mathbf{v}}{dt} = 0, & \mathbf{v}(t_0) = \eta \end{cases} \tag{3}$$

where $\mathbf{x} \equiv \mathbf{x}(t) \in \mathbb{R}^m$, $\mathbf{u} \equiv \mathbf{u}(t) \in \mathbb{R}^n$, and $\mathbf{v} \equiv \mathbf{v}(t) \in \mathbb{R}^n$, and $F(\cdot, \cdot, \mathbf{u}, \mathbf{v})$ characterizes the vector field and is parameterized by DNN with the conflicting sets of weights $(\mathbf{u}, \mathbf{v})$. Moreover, it is clear that (3) describes (2) wlog by taking $(\Phi, \ell) = (\mathcal{L}, 0)$. The functions $\Phi$, and $\ell$ are known as the terminal and running cost in the context of OCP, whereas in the DNN setting, they are equivalent to the loss function and the weight decay, respectively. Note that for this problem we consider symmetric layer-wise dynamics with respect to $\mathbf{u}, \mathbf{v}$. More explicitly, if a layer has a maximization counterpart, it has the same number of $\mathbf{u}$ and $\mathbf{v}$ parameters, as illustrated in the Node Level view of Figure 1.

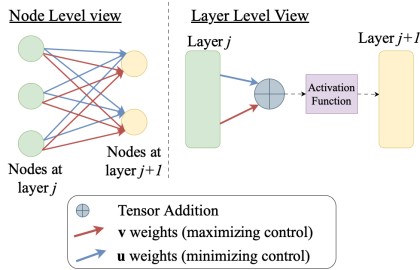

Figure 1: Node Level and Layer Level Architecture of Game Theoretic Layer

**Assumption 3.1.** The running cost $\ell$ is assumed to be quadratic with respect to the controls, more specifically: $\ell \equiv \ell(\mathbf{u}, \mathbf{v})$, with $\ell_{\mathbf{uu}} = R_{\mathbf{u}} I_n$, $\ell_{\mathbf{vv}} = -R_{\mathbf{v}} I_n$ and $\ell_{uv} = 0$, where $I_n \in \mathbb{R}^{n \times n}$ is the identity matrix, $R_{\mathbf{v}}$ and $R_{\mathbf{u}}$ are scalars, with $|R_{\mathbf{v}}| > |R_{\mathbf{u}}|$.

This assumption will enable us to write the Hessian in a form easily factorizable using Kronecker properties. The intuition for selecting $|R_{\mathbf{v}}| > |R_{\mathbf{u}}|$ is that we wish to penalize the disturbance more severely and prevent instability. The time-invariant ODEs imposed for $\mathbf{u}_t$, and $\mathbf{v}_t$ makes the ODE of $\mathbf{x}_t$ equivalent to (2), while also prompts us to interpret Problem (3) as the search for an optimal initial condition for time-invariant controls $\mathbf{u}_t$, and $\mathbf{v}_t$. Next, we define the accumulated loss $Q(t, \mathbf{x}, \mathbf{u}, \mathbf{v})$:

$$Q(t, \mathbf{x}, \mathbf{u}, \mathbf{v}) = \Phi(\mathbf{x}(t_f)) + \int_t^{t_f} \ell(\tau, \mathbf{x}(\tau), \mathbf{u}(\tau), \mathbf{v}(\tau)) d\tau \tag{4}$$

which implies that $0 = \ell(t, \mathbf{x}, \mathbf{u}, \mathbf{v}) + \frac{dQ(t,\mathbf{x},\mathbf{u},\mathbf{v})}{dt}$. At this point our goal is to compute higher-order derivatives with respect to $\mathbf{u}, \mathbf{v}$. Note that the first order derivative of $\mathcal{L}$ wrt to $\theta$, or $\eta$ under our OCP formulation is equivalent to $\frac{\partial Q}{\partial \mathbf{u}_{t_0}}$ or $\frac{\partial Q}{\partial \mathbf{v}_{t_0}}$ respectively, as $Q(t_0, \mathbf{x}_{t_0}, \mathbf{u}_{t_0}, \mathbf{v}_{t_0})$ is practically the accumulation of error on $[t_0, t_f]$. This explains our previous remark that in essence we search an optimal initial condition for $\mathbf{u}$, and $\mathbf{v}$. Now, we frame the equation above along a nominal path to derive the ODEs that will help us to compute the second order derivatives at $t_0$.

**Theorem 3.2.** *Consider Assumption 3.1, and a nominal trajectory $(\bar{\mathbf{x}}, \bar{\mathbf{u}}, \bar{\mathbf{v}})$, that satisfies the ODEs in the constraints of (3) . Then, the second order derivatives of $Q$ expanded locally around the nominal trajectory obey the following backward ODEs:*

$$-\frac{dQ_{\mathbf{xx}}}{dt} = F_{\mathbf{x}}Q_{\mathbf{xx}} + Q_{\mathbf{xx}}F_{\mathbf{x}}^{\mathsf{T}}, \quad -\frac{dQ_{\mathbf{xu}}}{dt} = F_{\mathbf{x}}Q_{\mathbf{xu}} + Q_{\mathbf{xx}}F_{\mathbf{u}}^{\mathsf{T}}, \quad -\frac{dQ_{\mathbf{uu}}}{dt} = \ell_{\mathbf{uu}} + F_{\mathbf{u}}Q_{\mathbf{xu}} + Q_{\mathbf{ux}}F_{\mathbf{u}}^{\mathsf{T}},$$

$$-\frac{dQ_{\mathbf{uv}}}{dt} = F_{\mathbf{u}}Q_{\mathbf{xv}} + Q_{\mathbf{ux}}F_{\mathbf{v}}^{\mathsf{T}}, \quad -\frac{dQ_{\mathbf{vv}}}{dt} = \ell_{\mathbf{vv}} + F_{\mathbf{v}}Q_{\mathbf{xv}} + Q_{\mathbf{vx}}F_{\mathbf{v}}^{\mathsf{T}}, \quad -\frac{dQ_{\mathbf{xv}}}{dt} = F_{\mathbf{x}}Q_{\mathbf{xv}} + Q_{\mathbf{xx}}F_{\mathbf{v}}^{\mathsf{T}}$$

$$-\frac{dQ_{\mathbf{u}}}{dt} = F_{\mathbf{u}}Q_{\mathbf{x}} + \ell_{\mathbf{u}}, \quad -\frac{dQ_{\mathbf{v}}}{dt} = F_{\mathbf{v}}Q_{\mathbf{x}} + \ell_{\mathbf{v}}$$

$$(5)$$

*The terminal conditions are given by: $Q_{\mathbf{x}}(t_f) = \Phi_{\mathbf{x}}, Q_{\mathbf{xx}}(t_f) = \Phi_{\mathbf{xx}}$, and $Q_{\mathbf{u}}(t_f) = \mathbf{0}_{n \times 1}, Q_{\mathbf{v}}(t_f) = Q_{\mathbf{uu}}(t_f) = Q_{\mathbf{vv}}(t_f) = Q_{\mathbf{uv}}(t_f) = \mathbf{0}_{n \times n}, Q_{\mathbf{xu}}(t_f) = Q_{\mathbf{xv}}(t_f) = \mathbf{0}_{n \times m}$.*

*Remark* 3.3. The proof for this Theorem is postponed for Appendix A.1 and it follows the standard derivation of continuous-time DDP, considering linearized ODE dynamics along the nominal trajectory path. This theorem implies that the proposed framework is based on min-max DDP, which has the favorable property of second order convergence and can obtain first and second order derivatives with a single pass of an ODE solver, without any recursive computations. This contributes to avoiding accumulating integration errors, and increased runtimes.

## 3.2 SECOND-ORDER MATRIX FACTORIZATION

The second order nature of our optimizer renders the efficient manipulation of second order terms a critical component of our study, as the amount of parameters in neural networks and neural ODEs grows easily to an unfavorable number.

**Theorem 3.4.** *We assume the matrix $Q_{xx}(t_f)$ to be a symmetric matrix of rank $R \leq m$, which may be represented as: $Q_{xx} = \sum_{i=1}^{R} \mathbf{y}_i \mathbf{y}_i^{\mathsf{T}}$, where $\mathbf{y}_i \in \mathbb{R}^m \; \forall t \in [t_0, t_f]$. Then let the second order matrices in (5) that contain derivative with respect to the state can be decomposed as follows:*

$$Q_{\mathbf{xx}}(t) = \sum_{i=1}^{R} \mathbf{q}_i(t)\mathbf{q}_i(t)^{\mathsf{T}}, \quad Q_{\mathbf{xu}}(t) = \sum_{i=1}^{R} \mathbf{q}_i(t)\mathbf{p}_i(t)^{\mathsf{T}}, \quad Q_{\mathbf{xv}}(t) = \sum_{i=1}^{R} \mathbf{q}_i(t)\mathbf{s}_i(t)^{\mathsf{T}} \quad (6)$$

*Then the vectors $\mathbf{q}_i \in \mathbb{R}^m$, and $\mathbf{p}_i(t)$, and $\mathbf{s}_i(t) \in \mathbb{R}^n$ obey the following backward ODEs:*

$$-\frac{d\mathbf{q}_i(t)}{dt} = F_{\mathbf{x}}\mathbf{q}_i(t), \quad -\frac{d\mathbf{p}_i(t)}{dt} = F_{\mathbf{u}}\mathbf{q}_i(t), \quad -\frac{d\mathbf{s}_i(t)}{dt} = F_{\mathbf{v}}\mathbf{q}_i(t) \quad (7)$$

*with the terminal condition given by $(\mathbf{q}_i(t_f), \mathbf{p}_i(t_f), \mathbf{s}_i(t_f)) = (\mathbf{y}_i, 0, 0)$. Additionally, the second order matrices $Q_{\mathbf{uu}}, Q_{\mathbf{uv}},$ and $Q_{\mathbf{vv}}$ can be represented as*

$$Q_{\mathbf{uu}}(t) = r(t) + \sum_{i=1}^{R} \mathbf{p}_i(t)\mathbf{p}_i(t)^{\mathsf{T}}, \; Q_{\mathbf{vv}}(t) = m(t) + \sum_{i=1}^{R} \mathbf{s}_i(t)\mathbf{s}_i(t)^{\mathsf{T}}, \; Q_{\mathbf{uv}}(t) = \sum_{i=1}^{R} \mathbf{p}_i(t)\mathbf{s}_i(t)^{\mathsf{T}},$$

$$(8)$$

*where $r(t) \equiv \ell_{\mathbf{uu}}(t_f - t)$ and $m(t) \equiv \ell_{\mathbf{vv}}(t_f - t)$.*

This theorem suggests that the coupled matrices back propagated through (5) can be decomposed into a set of vectors. For this reason, lower rank matrices observed in many Neural ODE applications alleviate the computational load and provide great memory efficiency (Chen et al., 2018). Substituting the ODEs from Eq. (7), into the expression in Eq. (8) yields:

$$Q_{\mathbf{uu}}(t) = R_{\mathbf{u}}I(t_f - t) + \sum_{i=1}^{R} \left( \int_{t_f}^{t} F_{\mathbf{u}}\mathbf{q}_i dt \right) \left( \int_{t_f}^{t} F_{\mathbf{u}}\mathbf{q}_i dt \right)^{\mathsf{T}} \quad (9)$$

Similar expressions for the other matrices are left in the Appendix A.2. To avoid dimensionality issues and tensor representations, the integrations in Eq. (9) are separated into each layer $j$ of the network. We denote $\mathbf{u}^j$, and $\mathbf{v}^j$ the parameters of layer $j$, and consider the preactivation vector $\mathbf{h}_j(t)$, as an affine combination of the weights with the input to the $j^{\text{th}}$ layer. Recall that we have considered symmetric dynamics layer-wise, implying that $F_{\mathbf{u}_j} = F_{\mathbf{v}_j}$, and in turn $\mathbf{h}_{\mathbf{u}}^j = \mathbf{h}_{\mathbf{v}}^j = \mathbf{z}_j$, and $\mathbf{h}_{\mathbf{x}}^j = (\mathbf{u} + \mathbf{v})$. This implies that $F_{u_j}\mathbf{q}_i = \mathbf{z}_j \otimes (\frac{\partial F}{\partial \mathbf{h}_j}\mathbf{q}_i)$ for feed-forward layers, and $F_{u_j}\mathbf{q}_i = \mathbf{z}_j \hat{*}(\frac{\partial F_{u_j}}{\partial \mathbf{h}_j}\mathbf{q}_i)$ for convolution layers. where $\otimes$ denotes the Kronecker product, and $\hat{*}$ denotes the de-convolution operator. Finally, assuming that $\mathbf{z}(t)_j$, and $\frac{\partial F}{\partial \mathbf{h}_j}$ are: i) uncorrelated across time, ii) and pair-wise independent, we can express the layer-wise precondition matrix $Q_{\mathbf{u}_j\mathbf{u}_j}$, at time $t = t_0$

$$Q_{\mathbf{u}_j\mathbf{u}_j}(t_0) \approx R_\mathbf{u} I(t_0 - t_f) + \int_{t_f}^{t_0} \underbrace{\left(\mathbf{z}_j\mathbf{z}_j^\intercal\right)\mathrm{dt}}_{A_j(t_0)} \otimes \int_{t_f}^{t_0} \underbrace{\sum_{i=1}^{R}\left(\left(\frac{\partial F}{\partial \mathbf{h}_j}\mathbf{q}_i\right)\left(\left(\frac{\partial F}{\partial \mathbf{h}_j}\mathbf{q}_i\right)\right)^\intercal\right)\mathrm{dt}}_{B_j(t_0)} . \tag{10}$$

Similarly for the other matrices: $Q_{\mathbf{v}_j\mathbf{v}_j}(t_0) = A_j(t_0) \otimes B_j(t_0) - R_\mathbf{v} I(t_0 - t_f)$, and $Q_{\mathbf{u}_j\mathbf{v}_j}(t_0) = A_j(t_0) \otimes B_j(t_0)$. Note that for the rest of this paper, we will drop the subscript relating to the $j^{\text{th}}$ layer, our analysis will still concern individual layer, but can be easily extended to the entire architecture. The detailed derivation along with a discussion over these assumptions takes place in Appendix A.4.

## 3.3 EFFICIENT DECOMPOSITION OF UPDATE LAWS

**Update Law from Min-Max DDP** Recall that Neural ODE analysis and OCP principles are deeply intertwined. That motivates us to view the optimization process of the Neural ODEs as a trajectory optimization task. Consider the closed loop Min-Max DDP update law

$$\begin{bmatrix} \delta\mathbf{u}_t \\ \delta\mathbf{v}_t \end{bmatrix} = \begin{bmatrix} l_\mathbf{u} \\ l_\mathbf{v} \end{bmatrix} + \begin{bmatrix} K_\mathbf{u} \\ K_\mathbf{v} \end{bmatrix} \delta\mathbf{x}_t, \tag{11}$$

where the feed-forward gains $l_\mathbf{u}$, along with $l_\mathbf{v}$, and the feedback gains $K_\mathbf{u}$, $K_\mathbf{v}$ are given by

$$\begin{aligned} l_\mathbf{u} &= \tilde{Q}_{\mathbf{uu}}^{-1}(Q_{\mathbf{uv}}Q_{\mathbf{vv}}^{-1}Q_\mathbf{v} - Q_\mathbf{u}), \text{ and } K_\mathbf{u} = \tilde{Q}_{\mathbf{uu}}^{-1}(Q_{\mathbf{ux}} - Q_{\mathbf{uv}}Q_{\mathbf{vv}}^{-1}Q_{\mathbf{vx}}) \\ l_\mathbf{v} &= \tilde{Q}_{\mathbf{vv}}^{-1}(Q_{\mathbf{uv}}Q_{\mathbf{uu}}^{-1}Q_\mathbf{u} - Q_\mathbf{v}), \text{ and } K_\mathbf{v} = \tilde{Q}_{\mathbf{vv}}^{-1}(Q_{\mathbf{vx}} - Q_{\mathbf{uv}}Q_{\mathbf{vv}}^{-1}Q_{\mathbf{vx}}) \end{aligned} \tag{12}$$

with $\tilde{Q}_{\mathbf{uu}} = (Q_{\mathbf{uu}} - Q_{\mathbf{uv}}Q_{\mathbf{vv}}^{-1}Q_{\mathbf{vu}})$, and $\tilde{Q}_{\mathbf{vv}} = (Q_{\mathbf{vv}} - Q_{\mathbf{vu}}Q_{\mathbf{uu}}^{-1}Q_{\mathbf{uv}})$. A detailed derivation of this update law is given in Appendix B.2. Setting $\delta\mathbf{x}_t = 0$ yields the open loop update scheme for the Min-Max DDP, which will be our focal point for this study. Notice that our method yields Gradient Descent Ascent (GDA; (Jin et al., 2020; Lin et al., 2020)) with Hessian Preconditioning for $Q_{\mathbf{uv}} = 0$, implying that open loop min-max DDP is essentially a generalization of GDA. In Appendix A.6, we also provide tractable expressions for the terms in the closed loop update rule.

*Remark* 3.5. To generate tractable expressions for the terms in Eq. (11), we leverage two basic properties of Kronecker products: **1.** $(A \otimes B + \lambda I)^{-1} = (U_A \otimes U_B)(\Sigma_A \otimes \Sigma_B + \lambda)^{-1}(U_A \otimes U_B)^\intercal$, **2.** $(A \otimes B)\text{vec}(\mathrm{X}) = \text{vec}(\mathrm{BXA}^\intercal)$.

**Proposition 3.6.** *From the eigenvalue decomposition of the Kronecker products of every matrix term in $\tilde{Q}_{\mathbf{uu}}, \tilde{Q}_{\mathbf{vv}}$, we derive these equivalent expression for their eigenvalue decomposition.*

$$\begin{aligned} \tilde{Q}_{\mathbf{uu}}^{-1} &= (U_A \otimes U_B)(S_\mathbf{u} - S_{\mathbf{uv}}S_\mathbf{v}^{-1}S_{\mathbf{uv}})^{-1}(U_A \otimes U_B)^\intercal \\ \tilde{Q}_{\mathbf{vv}}^{-1} &= (U_A \otimes U_B)(S_\mathbf{v} - S_{\mathbf{uv}}S_\mathbf{v}^{-1}S_{\mathbf{uv}})^{-1}(U_A \otimes U_B)^\intercal \end{aligned} \tag{13}$$

*where $S_\mathbf{u}, S_\mathbf{u}, S_{\mathbf{uv}}$ are the matrices containing the singular values of the matrix terms in $\tilde{Q}_{\mathbf{uu}}, \tilde{Q}_{\mathbf{vv}}$.*

**Proposition 3.7.** *Following Property 3.6 and leveraging the second property mentioned in Remark 3.5 we can rewrite the feed forward gains in 43 more compactly*

$$l_\mathbf{u} = vec(U_B G_{\mathbf{uv}} U_A^\intercal) - vec(U_B G_{\mathbf{uu}} U_A^\intercal), \quad l_\mathbf{v} = vec(U_B G_{\mathbf{vu}} U_A^\intercal) - vec(U_B G_{\mathbf{vv}} U_A^\intercal) \tag{14}$$

*where $G_{\mathbf{uv}}, G_{\mathbf{uv}}, G_{\mathbf{uv}}, G_{\mathbf{uv}}$ are matrix terms defined in the Appendix A.6.*

Detailed proofs for Propositions 3.6, and 3.7 are left in the Appendix A.5, and A.6 respectively. Algorithm 1 summarizes our Game Theoretic optimizer.

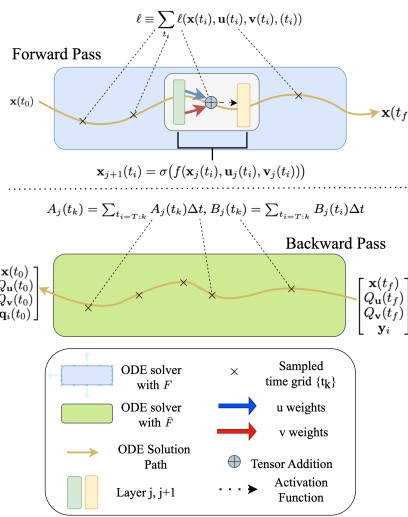

---

**Algorithm 1** GTSONO

1: **Input**: dataset $\mathcal{D}$, parameterized vector field $F(\cdot, \cdot, \mathbf{u}, \mathbf{v})$, integration time $[t_0, t_f]$, ODESolver: 'ODESolve', learning rate $\eta$, time step $\Delta t$, Tikhonov regularization constants $R_\mathbf{u}, R_\mathbf{v}$
2: **repeat**
3:   $\mathbf{x}(t_f) = ODESolve(\mathbf{x}(t_0), t_0, t_f, F)$, where $x(t_0) \sim \mathcal{D}$
4:   Initialize Optimizer
5:   **for** $t_i$ in $\{t_f, t_f + \Delta t, \ldots, t_0 + \Delta t, t_0\}$
6:     $[\mathbf{x}(t_{i-1}, Q_\mathbf{u}(t_{i-1}), Q_\mathbf{v}(t_{i-1}), \mathbf{q}_i(t_{i-1})] = $
        $ODESolve([\mathbf{x}(t_i), Q_\mathbf{u}(t_i), Q_\mathbf{v}(t_i), \mathbf{q}_i(t_i)],$
        $t_{i-1}, t_i, \bar{F})$
7:     For layer $j$ evaluate $A_j(t_i)$, and $B_j(t_i)$
8:   **end for**
9:   $A_j(t_0) = \sum_{t_i} A_j(t_i)\Delta t, B_j(t_0) = \sum_{t_i} B_j(t_i)\Delta t$
10:   Compute $\ell_\mathbf{u}, \ell_\mathbf{v}$ from Eq. 14
11:   Update controls: $\mathbf{u} \leftarrow \mathbf{u} + \eta\ell_\mathbf{u}, \mathbf{v} \leftarrow \mathbf{v} + \eta\ell_\mathbf{v}$
12: **until** converges

---

Figure 2: GTSONO overview, where F: forward dynamics F, and $\bar{F}$: backward dynamics

## 4 CONVERGENCE

In *sequential* two-player zero sum games, a global minimax point always exists even if $f$ is nonconvex-nonconcave, due to the extreme-value theorem (Jin et al. (2020)). Conversely, granted the updates of the proposed algorithm take place *simultaneously*, and due to the non-convexity of problem (3) finding a global saddle point is NP-hard, thus we settle for local saddle points. However, even that task is not straightforward, as in the simultaneous min-max setting, GDA methods may display some undesirable behaviours, such as convergence to a non-critical point (Daskalakis & Panageas, 2018), or to an unstable local saddle point (Wang et al., 2019). In this regard, we provide local convergence guarantees for our optimizer to a locally stable saddle point.

For convenience on handling the variables we further define the vector $\mathbf{z} = [\mathbf{u}; \mathbf{v}] \in \mathbb{R}^{2n}$, along with the function $G : \mathbb{R}^{2n} \to \mathbb{R}^{2n}$, as $G(\mathbf{z}) = [\nabla_\mathbf{u} Q(\mathbf{u}, \mathbf{v}); -\nabla_\mathbf{v} Q(\mathbf{u}, \mathbf{v})]$. Accordingly the Jacobian of G is given by: $JG(\mathbf{z}) = \begin{bmatrix} Q_\mathbf{uu} & Q_\mathbf{uv} \\ -Q_\mathbf{vu} & -Q_\mathbf{vv} \end{bmatrix} \in \mathbb{R}^{2n \times 2n}$. Notice that after computing the inverse of $JG(\mathbf{z})$ using the Schur complement (Liu et al., 2021b), the preconditioned update rule can be rewritten as $\mathbf{z}_{k+1} = \mathbf{z}_k - \eta JG^{-1}(\mathbf{z}_k)G(\mathbf{z}_k)$ yielding the update rule for the open loop Min-Max DDP shown in Eq. 11, for $\delta\mathbf{x}_t = 0$. Let us start with the definition and sufficient conditions of a local saddle point.

**Definition 4.1.** Let $K_\gamma = \{\mathbf{z} \mid \|\mathbf{z} - \mathbf{z}^\star\| \le \gamma\}$, be a neighborhood around fixed point $\mathbf{z}^\star = [\mathbf{u}^\star; \mathbf{v}^\star]$. Then, $\mathbf{z}^\star$ is a local saddle point, if $f(\mathbf{u}^\star, \mathbf{v}) \le f(\mathbf{u}^\star, \mathbf{v}^\star) \le f(\mathbf{u}, \mathbf{v}^\star), \forall \mathbf{u}, \mathbf{v} \in K_\gamma$.

**Assumption 4.2.** Let $\mathbf{z}^\star = [\mathbf{u}^\star; \mathbf{v}^\star]$ be a stationary point. Recall that we have $Q_\mathbf{uu} = A \otimes B + R_\mathbf{u}I$, and $Q_\mathbf{vv} = A \otimes B - R_\mathbf{v}I$, for $R_\mathbf{u}, R_\mathbf{v} > 0$. Assuming that wlog $\lambda_{\min}(A \otimes B) \ge -m$, and $\lambda_{\max}(A \otimes B) \le M$, for $m, M > 0$, then there exist appropriate $R_\mathbf{u} > 0$ and $R_\mathbf{v} > 0$ large enough such that $\lambda_{\min}(Q_\mathbf{uu}) > 0$, and $\lambda_{\max}(Q_\mathbf{vv}) < 0$, ensuring that $\mathbf{z}^\star$ is a local saddle point.

Our next step is to prove local convergence guarantees to a local saddle point $\mathbf{z}^\star$.

**Theorem 4.3.** *Suppose $\mathbf{z}_0 \in K_\gamma$. Then for $L$-Lipschitz function $Q(\mathbf{z})$, with also $L'$-Lipschitz Jacobian of $G(\mathbf{z})$, the accumulative difference between iterates and fixed point $\mathbf{z}^\star$, as well as the total error from $Q(\mathbf{z}^\star)$ can be bounded by a constant*

$$\sum_{k=0}^{K} \|\mathbf{z}_k - \mathbf{z}^\star\| \le O(1), \qquad \sum_{k=0}^{K} |Q(\mathbf{z}_k) - Q(\mathbf{z}^\star)| \le O(1). \qquad (15)$$

This implies that the iterates converge and the total error goes to 0, as $K \to \infty$. The proof of Theorem 4.3 can be found in Appendix C.1. Finally, we proceed to show that our algorithm converges to a strictly stable fixed pointed, implying local convergence (Wang et al. (2019)).

**Lemma 4.4** (Proposition 1.4 (Daskalakis & Panageas, 2018)). *Assume fixed point $z^\star = (\mathbf{u}^\star, \mathbf{v}^\star)$, for which it holds $\nabla f(z^\star) = 0$ is locally stable if the Jacobian of the update rule satisfies $\rho(J) \leq 1$, where $\rho(\cdot)$ is the spectral norm.*

To leverage the Lemma above, we begin with the Jacobian of our update rule

$$J = I - \eta(JG(\mathbf{z}_k)^{-1} JG(\mathbf{z}_k) + \nabla_{\mathbf{z}_k}(JG^{-1}(\mathbf{z}_k))G(\mathbf{z}_k)) \tag{16}$$

**Proposition 4.5.** *Considering linearized ODE dynamics, we can deduce that for the derivatives $Q_{\mathbf{uu}}$ and $Q_{\mathbf{vv}}$, with respect to $\mathbf{z}_k = [\mathbf{u}_k, \mathbf{v}_k]$: $\frac{\partial}{\partial \mathbf{z}_k} Q_{\mathbf{uu}} = \frac{\partial}{\partial \mathbf{z}_k} Q_{\mathbf{vv}} = 0$, implying $\nabla(JG^{-1}(\mathbf{z}_k)) = 0$.*

From this proposition, we can easily infer for Eq. (16) that $||J|| < 1, \forall \eta \in (0, 1)$. Therefore, our optimizer convergenges to stable saddle points, implying stability. The proof of Proposition 4.5 has been postponed for Appendix C.2.

## 5 EXPERIMENTS

We validate the efficacy of our algorithm in comparison to other state-of-the-art optimizers widely used in Neural ODE applications, and we highlight the ability of GTSONO to be successfully adapted to adversarial training methods. All experiments are conducted on a TITAN RTX.

**Datasets** We carry out our experiments on two image datasets: CIFAR and SVHN. Both datasets have been standardized and consist of $3\times32\times32$ colour images, and 10 label classes.

**Networks** The benchmark optimizers were applied on identical network structures, for every experiment round. GTSONO was also evaluated on the same formation with two identical sets of antagonizing weights, implying double parameters compared to the benchmark models. Furthermore, to reduce the number of trainable parameters, we also consider a framework containing the antagonizing set of controls only in the convolution layers. We will refer to this architecture as *c-GTSONO*. More information regarding the network architectures are provided in the Appendix D.

**Attacks** In this study, we focus on the white box $\ell_\infty$-norm PGD, FGSM attacks and gray/ black-box attack CW attack to assess the robustness of our models. Every disturbance mentioned in our experiments below is applied on the standardized dataset. Details about the update rules of the attacks are left for Appendix D. We denote each model's accuracy to the clean test set (i.e. natural accuracy) with $A_{nat}$; $PGD_s^\epsilon$ the PGD attack with a perturbation distance $\epsilon$ that takes $s$ steps in the direction of the gradient; and $FGSM^\alpha$ the FGSM attack whose single step in the direction of the gradient is multiplied with constant $\alpha$. The CW is denoted simply as $CW_\infty$ as the $\ell_\infty$ norm was considered and for every experiment the pertubation distance was set to $\epsilon = 0.03$, and max-iterations $K = 100$.

### 5.1 RESULTS

**Optimizer Comparison** For this round of experimens we compare GTSONO with Adam SGD with momentum as first order benchmark optimizers, and SNOpt (Liu et al., 2021a) as a second order baseline, on the two aforementioned datasets. The training for all optimizers was carried out using only non-perturbed images, with a batch size of 500 images on every dataset. The networks trained on both datasets was trained for 15 epochs. More details about the structure the network of each optimizer are provided in the Appendix D.

As shown from Tables 1 and 2, GTSONO outperforms the benchmark optimizers in both datasets with natural training, and that the performance gap increases as the degree of perturbation also increases. Additionally, it is also shown that GTSONO provides more robust predictions on average, as well as more confident as the standard deviation is smaller compared to the benchmark optimizers. Finally, we observe that the C-GTSONO version of our algorithm is the most efficient and best performing providing more accurate predictions in less training time. More details about the training times and memory consumption of each optimizer are left in the Appendix D.

**GTSONO on Adversarial Training Methods** In this round of experiments, we highlight the applicability of GTSONO to be efficiently adapted to other adversarial training methods and enhance

their performance. More specifically, we consider the Free Adversarial Training (FreeAT) scheme (Shafahi et al., 2019)) and TRADES (Zhang et al., 2019b), with SGD as their optimizer, and evaluate how these adversarial training methods benefit from employing C-GTSONO as their optimizer, instead of SGD. We perform two ablation studies by experimenting with the number of examined internal iterations for FreeAT are $m = 4, 8$, while in TRADES we fix the number of internal iterations to generate adversarial examples to 5 and evaluated for $1/\lambda = 6, 10$. More details about each adversarial training method and the selection of their hyperparameters are left in Appendix D.

Table 1: Average $\pm$ standard deviation of test set accuracy (%) on the CIFAR10 for each optimizer. $A_{nat}$ denotes the natural accuracy. $PGD_s^\epsilon$ denotes the accuracy under PGD attack, taking $s$ steps in the direction of the gradient with a perturbation distance $\epsilon$. $FGSM_\alpha$ describes the accuracy under FGSM attack where the single gradient step is multiplied with constant $\alpha$. $CW_\infty$ denotes the accuracy under the CW attack.

| Optimizer | $A_{nat}$ | $FGSM_{0.03}$ | $FGSM_{0.05}$ | $PGD_{0.03}^{20}$ | $PGD_{0.05}^{20}$ | $CW_\infty$ |
|---|---|---|---|---|---|---|
| Adam | $78.7 \pm 1.1$ | $48.2 \pm 0.7$ | $30.8 \pm 0.5$ | $45.1 \pm 1.2$ | $29.1 \pm 0.3$ | $15.4 \pm 0.6$ |
| SGD | $77.5 \pm 0.6$ | $47.3 \pm 1.3$ | $33.8 \pm 1.3$ | $45.8 \pm 1.5$ | $29.3 \pm 1.4$ | $18.3 \pm 1.6$ |
| SNOpt | $\mathbf{79.1 \pm 0.4}$ | $48.7 \pm 1.0$ | $35.7 \pm 1.0$ | $46.8 \pm 1.4$ | $32.1 \pm 1.4$ | $7.5 \pm 1.0$ |
| *GTSONO* | $74.7 \pm 0.6$ | $51.7 \pm \mathbf{0.3}$ | $37.9 \pm 0.4$ | $49.9 \pm 0.5$ | $34.9 + 1.1$ | $18.0 \pm 1.5$ |
| *C-GTSONO* | $74.7 \pm 0.7$ | $\mathbf{51.8} \pm 0.4$ | $\mathbf{38.0 \pm 0.2}$ | $\mathbf{50.6 \pm 0.3}$ | $\mathbf{35.0 + 0.2}$ | $\mathbf{36.3 \pm 2.2}$ |

Table 2: Average $\pm$ standard deviation of test set accuracy (%) on the SVHN for each optimizer.

| Optimizer | $A_{nat}$ | $FGSM_{0.03}$ | $FGSM_{0.05}$ | $PGD_{0.03}^{20}$ | $PGD_{0.05}^{20}$ | $CW_\infty$ |
|---|---|---|---|---|---|---|
| Adam | $98.9 \pm 0.3$ | $73.8 \pm 0.4$ | $55.9 \pm 0.8$ | $71.8 \pm \mathbf{0.1}$ | $48.4 \pm 1.0$ | $20.3 \pm 0.2$ |
| SGD | $98.4 \pm 0.0$ | $74.4 \pm 0.4$ | $56.1 \pm 0.5$ | $72.4 \pm 0.7$ | $50.4 \pm 1.1$ | $23.4 \pm 0.9$ |
| SNOpt | $99.1 \pm 0.1$ | $73.5 \pm 2.2$ | $54.4 \pm 2.3$ | $71.9 \pm 2.7$ | $48.7 \pm 3.3$ | $22.4 \pm 0.9$ |
| *GTSONO* | $\mathbf{99.6 \pm 0.0}$ | $78.0 \pm 0.4$ | $58.9 \pm 0.4$ | $76.7 \pm 0.8$ | $54.3 \pm 0.8$ | $31.6 \pm 2.2$ |
| *C-GTSONO* | $97.3 \pm 0.2$ | $\mathbf{80.8 \pm 0.2}$ | $\mathbf{65.2 \pm 0.3}$ | $\mathbf{80.3} \pm 0.4$ | $\mathbf{62.3 \pm 0.3}$ | $\mathbf{50.5 \pm 0.8}$ |

Table 3: Comparison between GTSONO and SGD applied on FreeAT, for $m = 4, 8$.

| Optimizer | $A_{nat}$ | $PGD_{20}^{0.03}$ | $PGD_{40}^{0.03}$ | $CW_\infty$ |
|---|---|---|---|---|
| SGD (m=4) | $\mathbf{76.8}$ | 48.8 | 48.1 | 6.5 |
| *C-GTSONO* (m=4) | 75.7 | $\mathbf{50.5}$ | $\mathbf{50.2}$ | $\mathbf{17.4}$ |
| SGD (m=8) | $\mathbf{76.2}$ | 54.8 | 54.2 | 9.5 |
| *C-GTSONO* (m=8) | 74.4 | $\mathbf{56.5}$ | $\mathbf{56.4}$ | $\mathbf{18.9}$ |

To avoid robust overfitting, we evaluate the robustness of each optimizer, after convergence is observed. Although GTSONO presents a slower per-iteration training time, from Figure 3 we observe that in both methods our optimizer converges to a local saddle point in considerable fewer iterations, due to the higher order convergence rate inherited by min-max DDP. In both defense schemes, GTSONO was found to converge approximately after 10 or less epochs, whereas the benchmark models required approximately double epochs to converge. This results in less training time overall, which is critical in adversarial training methods which generally take longer to train due to the internal iterations required to generate adversarial samples. Table 5 present a comparison between the training time required by GTSONO and the baseline model in FreeAT for $m = 8$, demonstrating the faster wall clock adversarial training time of GTSONO. The remaining comparisons about the required resources are left in the Appendix D. Finally, Tables 3 and 4 indicate that the adaptation of our optimizer increases the robust accuracy in both defense methods. More specifically, FreeAT benefited by the adaptation of GTSONO by a performance increase of approximately 2% in PGD attacks, whereas in TRADES we obtained a performance increase of more than 4% for both tested values of $\lambda$. Interestingly, the benchmark models were found to perform better under $CW_\infty$.

**Update Laws** Recall, that in this study our focal point was the open-loop Min-Max DDP, however as mentioned in in Section 3, setting the cross-terms: $Q_{\mathbf{vu}} = 0$, we readily obtain GDA with Hessian Preconditioning. We investigate the difference in the performance offered by the open loop update rule

Table 4: Comparison between GTSONO and SGD used with TRADES, for $\lambda^{-1} = 6, 10$.

| Method | $A_{nat}$ | $PGD_{20}^{0.03}$ | $PGD_{40}^{0.03}$ | $CW_\infty$ |
|---|---|---|---|---|
| TRADES($\lambda^{-1}$=6) | 73.2 | 54.5 | 54.3 | **22.8** |
| *C-GTSONO*($\lambda^{-1}$=6) | **75.8** | **58.9** | **58.8** | 20.4 |
| TRADES($\lambda^{-1}$=10) | 69.4 | 57.2 | 56.9 | **25.2** |
| *C-GTSONO*($\lambda^{-1}$=10) | **72.6** | **61.2** | **61.0** | 20.7 |

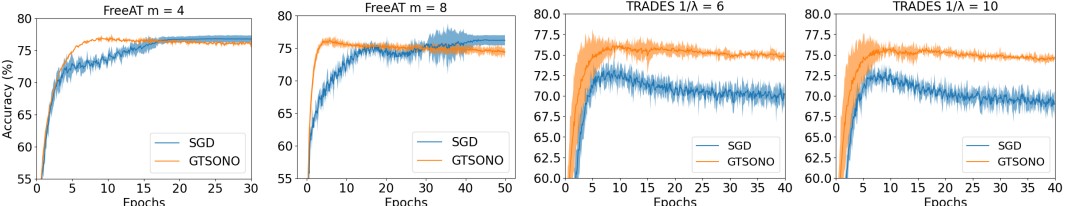

Figure 3: Convergence comparison between GTSONO and SGD in *Left* : FreeAT, *Right* : TRADES

(*DDP-GTSONO*), and by the GDA update (*GDA-GTSONO*), under the FGSM, PGD for increasing perturbation. In Figure 4 we observe that on the SVHN dataset, DDP-GTSONO outperforms its counterpart, for every examined disturbance. Additionally on the CIFAR 10 dataset, GDA-GTSONO is more accurate for small perturbations, however for increasing disturbance, it is observed that DDP-GTSONO achieves better robust accuracy against both attacks, demonstrating the superiority of our DDP based algorithm.

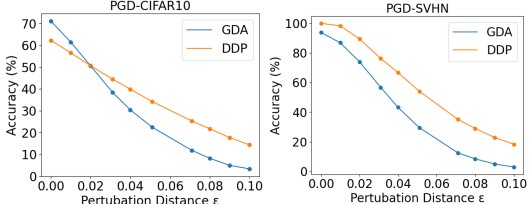

Figure 4: Comparison of DDP and GDA GTSONO

Table 5: Training time per iterations and epochs trained for FreeAT, $m = 4$ (min:sec)

| Optimizer | Time per iteration (sec) | Epochs |
|---|---|---|
| SGD | **0.63** | 40 |
| *C-GTSONO* | 1.4 | **5** |

**Limitations** The achieved robustness of our optimizer comes at the price of a slower per iteration training time and higher memory consumption, mainly due to the required computation of $Q_{\mathbf{uu}}, Q_{\mathbf{uv}}, Q_{\mathbf{vv}}$. However, as shown in Appendix D, it is suggested that GTSONO is still affordable for current GPUs. Additionally, we observed that the continuous models considered for this study were found to be prone to robust overfitting, in some instances not only in our algorithm but also in the benchmark optimizers as well.

## 6 CONCLUSIONS

We present an efficient game theoretic optimizer for training robust neural ODEs, with provable convergence guarantees to local saddle points. Based on the principles of min-max DDP, our framework leverages the second order convergence of this OC paradigm by employing Kronecker factorization to decompose matrices into vectors which can be easily obtained through solving the ODE. This allows us to derive tractable expressions for the terms in the update of open loop min-max DDP. Empirically, GTSONO increased the robustness and generated more accurate and confident predictions on attacked images compared to benchmark optimizers under natural training. We also demonstrated that GTSONO is successfully adapted to already existing adversarial training methods, enhancing their robustness. In future work, we wish to extend the applicability of our optimizer to a larger variety of network architectures (e.g. ResNets, Transformers), and explore appropriate regularization techniques, to further improve the performance of our algorithm. Finally, we wish to investigate recasting the proposed optimizer through distributed DDP schemes, as in (Saravanos et al., 2023). In summary, our work paves new ways for robust optimal control methodologies to be applied in deep continuous models to enhance their robustness.

ACKNOWLEDGEMENTS

This material is based upon work supported by the NASA Aeronautics Research Mission Directorate (ARMD) University Leadership Initiative (ULI) under cooperative agreement number 80NSSC22M0070. Augustinos Saravanos acknowledges financial support by the A. Onassis Foundation Scholarship.

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

## A   MISSING DERIVATIONS FROM SECTION 3

We recall the formulation of the objective of the Neural ODE framework expressd in a fashion easily interpretable through the prism of game theoretic OCP.

$$\min_{\mathbf{u}} \max_{\mathbf{v}} \left[ \Phi(x_{t_f}) + \int_{t_0}^{t_f} \ell(t, \mathbf{x}, \mathbf{u}, \mathbf{v}) dt \right], \text{subjected to} \begin{cases} \frac{d\mathbf{x}}{dt} = F(t, \mathbf{x}, \mathbf{u}, \mathbf{v}), & \mathbf{x}(t_0) = x_0 \\ \frac{d\mathbf{u}}{dt} = 0, & \mathbf{u}(t_0) = \theta \\ \frac{d\mathbf{v}}{dt} = 0, & \mathbf{v}(t_0) = \eta \end{cases}$$

(17)

where $\mathbf{x} \equiv \mathbf{x}(t) \in \mathbb{R}^m$, $\mathbf{u} \equiv \mathbf{u}(t) \in \mathbb{R}^n$, and $\mathbf{v} \equiv \mathbf{v}(t) \in \mathbb{R}^n$. It is clear that (17) describes (2) without loss of generality by taking $(\Phi, \ell) = (\ell, 0)$. The function $F(t, \mathbf{x}, \mathbf{u}, \mathbf{v})$ characterizes the vector field and is parameterized by a Deep Neural Network (DNN). We consider that the dynamics of the system are symetric with respect to the two sets of weights $(\mathbf{u}, \mathbf{v})$. The functions $\Phi$, and $\ell$ are known as the terminal and running cost in the context of OCP. This problem is understood as particular type of OCP that searches for the optimal initial condition of the time-invariant controls $\mathbf{u}_t, \mathbf{v}_t$. In the DNN setting, the terminal cost is equivalent to the loss function, for instance Categorical Cross Entropy for multi-label classification, and the running cost is equivalent to the weight decay, or some other regularization techniques that acts on the weights of the intermediate hidden layers. Next, the accumulated loss $Q(t, \mathbf{x}, \mathbf{u}, \mathbf{v})$ is defined as follows:

$$Q(t, \mathbf{x}, \mathbf{u}, \mathbf{v}) = \Phi(\mathbf{x}(t_f)) + \int_t^{t_f} \ell(\tau, \mathbf{x}(\tau), \mathbf{u}(\tau), \mathbf{v}(\tau)) d\tau$$

(18)

From this definition of the accumulated loss implies that $Q$ we can readily obtain:

$$0 = \ell(t, \mathbf{x}, \mathbf{u}, \mathbf{v}) + \frac{dQ(t, \mathbf{x}, \mathbf{u}, \mathbf{v})}{dt}$$

(19)

### A.1   PROOF OF THEOREM 3.2

Considering the nominal trajectory $(\bar{\mathbf{x}}, \bar{\mathbf{u}}, \bar{\mathbf{v}})$, that satisfies the ODEs in the constraints of (3), we take the Taylor expansion of the terms in Eq.19 keeping up to second order terms:

$$\ell(\mathbf{x}, \mathbf{u}, \mathbf{v}) = \ell(\bar{\mathbf{x}}, \bar{\mathbf{u}}, \bar{\mathbf{v}}) + \ell_{\mathbf{x}} \delta\mathbf{x}_t + \ell_{\mathbf{u}} \delta\mathbf{u} + \ell_{\mathbf{v}} \delta\mathbf{v} + \frac{1}{2} \begin{bmatrix} \delta\mathbf{x}_t \\ \delta\mathbf{u}_t \\ \delta\mathbf{v}_t \end{bmatrix}^{\mathsf{T}} \begin{bmatrix} \ell_{xx} & \ell_{xu} & \ell_{xv} \\ \ell_{ux} & \ell_{uu} & \ell_{uv} \\ \ell_{vx} & \ell_{vu} & \ell_{vv} \end{bmatrix} \begin{bmatrix} \delta\mathbf{x}_t \\ \delta\mathbf{u}_t \\ \delta\mathbf{v}_t \end{bmatrix} \quad (20a)$$

$$Q(\mathbf{x}, \mathbf{u}, \mathbf{v}) = Q(\bar{\mathbf{x}}, \bar{\mathbf{u}}, \bar{\mathbf{v}}) + Q_{\mathbf{x}} \delta\mathbf{x}_t + Q_{\mathbf{u}} \delta\mathbf{u} + Q_{\mathbf{v}} \delta\mathbf{v} + \frac{1}{2} \begin{bmatrix} \delta\mathbf{x}_t \\ \delta\mathbf{u}_t \\ \delta\mathbf{v}_t \end{bmatrix}^{\mathsf{T}} \begin{bmatrix} Q_{xx} & Q_{xu} & Q_{xv} \\ Q_{ux} & Q_{uu} & Q_{uv} \\ Q_{vx} & Q_{vu} & Q_{vv} \end{bmatrix} \begin{bmatrix} \delta\mathbf{x}_t \\ \delta\mathbf{u}_t \\ \delta\mathbf{v}_t \end{bmatrix}$$

(20b)

We differentiate 20b, in order to utilize Eq. 19, and taking into account that $\frac{d\delta\mathbf{u}}{dt} = \frac{\delta\mathbf{v}}{dt} = 0$, we obtain:

$$\frac{dQ}{dt} \approx \frac{dQ(\bar{\mathbf{x}}, \bar{\mathbf{u}}, \bar{\mathbf{v}})}{dt} + \left( \frac{dQ_{\mathbf{x}}}{dt} \delta\mathbf{x}_t + Q_{\mathbf{x}} \frac{d\delta\mathbf{x}}{dt} \right) + \left( \frac{dQ_{\mathbf{u}}}{dt} \delta\mathbf{u} + Q_{\mathbf{u}} \frac{d\delta\mathbf{u}}{dt} \right) + \left( \frac{dQ_{\mathbf{v}}}{dt} \delta\mathbf{v} + Q_{\mathbf{v}} \frac{d\delta\mathbf{v}}{dt} \right) +$$

$$+ \frac{1}{2} \begin{bmatrix} \delta\mathbf{x}_t \\ \delta\mathbf{u}_t \\ \delta\mathbf{v}_t \end{bmatrix}^{\mathsf{T}} \begin{bmatrix} \frac{dQ_{xx}}{dt} & \frac{dQ_{xu}}{dt} & \frac{Q_{xv}}{dt} \\ \frac{dQ_{ux}}{dt} & \frac{Q_{uu}}{dt} & \frac{Q_{uv}}{dt} \\ \frac{dQ_{vx}}{dt} & \frac{Q_{vu}}{dt} & \frac{Q_{vv}}{dt} \end{bmatrix} \begin{bmatrix} \delta\mathbf{x}_t \\ \delta\mathbf{u}_t \\ \delta\mathbf{v}_t \end{bmatrix} + \frac{1}{2} \begin{bmatrix} \frac{\delta\mathbf{x}_t}{dt} \\ 0 \\ 0 \end{bmatrix}^{\mathsf{T}} \begin{bmatrix} Q_{xx} & Q_{xu} & Q_{xv} \\ Q_{ux} & Q_{uu} & Q_{uv} \\ Q_{vx} & Q_{vu} & Q_{vv} \end{bmatrix} \begin{bmatrix} \delta\mathbf{x}_t \\ \delta\mathbf{u}_t \\ \delta\mathbf{v}_t \end{bmatrix}$$

$$+ \frac{1}{2} \begin{bmatrix} \delta\mathbf{x}_t \\ \delta\mathbf{u}_t \\ \delta\mathbf{v}_t \end{bmatrix}^{\mathsf{T}} \begin{bmatrix} Q_{xx} & Q_{xu} & Q_{xv} \\ Q_{ux} & Q_{uu} & Q_{uv} \\ Q_{vx} & Q_{vu} & Q_{vv} \end{bmatrix} \begin{bmatrix} \frac{\delta\mathbf{x}_t}{dt} \\ 0 \\ 0 \end{bmatrix}$$

(21)

Next, we need to compute the term $\frac{\delta\mathbf{x}}{dt}$:

$$\frac{d\delta\mathbf{x}}{dt} = \frac{d\mathbf{x}}{dt} - \frac{\bar{\mathbf{x}}}{dt} = F_{\mathbf{x}} \delta\mathbf{x} + F_{\mathbf{u}} \delta\mathbf{u} + F_{\mathbf{v}} \delta\mathbf{v}$$

(22)

The second equation comes from the fact that the time derivative evaluated for fixed $\bar{\mathbf{x}}$ yields the dynamics $F(\bar{\mathbf{x}}, \bar{\mathbf{u}}, \bar{\mathbf{v}})$, whereas for the first time derivative we take the Taylor expansion around the

nominal trajectory. Finally, substituting Eq. 22 into Eq. 21, and back into 19, we obtain the following ODE for Q and its derivatives

$$-\frac{dQ_{\mathbf{x}}}{dt} = F_{\mathbf{x}}Q_{\mathbf{x}} + \ell_{\mathbf{x}}, \quad -\frac{dQ_{xx}}{dt} = \ell_{xx} + F_x Q_{xx} + Q_{\mathbf{xx}}F_{\mathbf{x}}^{\mathsf{T}}, \quad -\frac{dQ_{\mathbf{xu}}}{dt} = \ell_{\mathbf{xu}} + F_x Q_{\mathbf{xu}} + Q_{\mathbf{xx}}F_{\mathbf{u}}^{\mathsf{T}}$$

$$(23a)$$

$$-\frac{dQ_{\mathbf{u}}}{dt} = F_{\mathbf{u}}Q_{\mathbf{x}} + \ell_{\mathbf{u}}, \quad -\frac{dQ_{\mathbf{uu}}}{dt} = \ell_{\mathbf{uu}} + F_{\mathbf{u}}Q_{\mathbf{xu}} + Q_{\mathbf{ux}}F_{\mathbf{u}}^{\mathsf{T}}, \quad -\frac{dQ_{\mathbf{uv}}}{dt} = \ell_{\mathbf{uv}} + F_{\mathbf{u}}Q_{\mathbf{xv}} + Q_{\mathbf{ux}}F_{\mathbf{v}}^{\mathsf{T}}$$

$$(23b)$$

$$-\frac{dQ_{\mathbf{v}}}{dt} = F_{\mathbf{v}}Q_{\mathbf{x}} + \ell_{\mathbf{v}}, \quad -\frac{dQ_{\mathbf{vv}}}{dt} = \ell_{\mathbf{vv}} + F_{\mathbf{v}}Q_{\mathbf{xv}} + Q_{\mathbf{vx}}F_{\mathbf{v}}^{\mathsf{T}}, \quad -\frac{dQ_{\mathbf{xv}}}{dt} = \ell_{\mathbf{xv}} + F_x Q_{\mathbf{xv}} + Q_{\mathbf{xx}}F_{\mathbf{v}}^{\mathsf{T}}$$

$$(23c)$$

At this point, recall Assumption 3.1, then we easily obtain Eq. 5 for the ODEs of the second order derivatives.

## A.2 PROOF THEOREM 3.4

*Proof.* We recall 3.4, where it was assumed the matrix $Q_{xx}(t_1)$ to be a symmetric matrix of rank $R \leq m$, it may be represented as: $Q_{xx} = \sum_{i=1}^{R} \mathbf{y}_i \mathbf{y}_i^{\mathsf{T}}$, where $\mathbf{y}_i \in \mathbb{R}^m$. Additionally, we had that $Q_{uu}(t_f) = 0$, $Q_{vv}(t_f) = 0$. Then $\forall t \in [t_0, t_f]$, the second order matrices in (23) that contain derivative with respect to the state can be decomposed as follows:

$$Q_{\mathbf{xx}}(t) = \sum_{i=1}^{R} \mathbf{q}_i(t)\mathbf{q}_i(t)^{\mathsf{T}}, \quad Q_{\mathbf{xu}}(t) = \sum_{i=1}^{R} \mathbf{q}_i(t)\mathbf{p}_i(t)^{\mathsf{T}}, \quad Q_{\mathbf{xv}}(t) = \sum_{i=1}^{R} \mathbf{q}_i(t)\mathbf{s}_i(t)^{\mathsf{T}} \quad (24)$$

From Eq. 23, we obtain:

$$-\frac{dQ_{\mathbf{xx}}}{dt} = \ell_{\mathbf{xx}} + F_x Q_{xx} + Q_{xx}F_{\mathbf{x}}^{\mathsf{T}}$$

$$= F_x \sum_{i=1}^{R} \mathbf{q}_i(t)\mathbf{q}_i(t)^{\mathsf{T}} + \sum_{i=1}^{R} \Big(\mathbf{q}_i(t)\mathbf{q}_i(t)^{\mathsf{T}}\Big)F_{\mathbf{x}}^{\mathsf{T}} \quad (25)$$

$$= \sum_{i=1}^{R} \Big(F_x \mathbf{q}_i(t)\Big)\mathbf{q}_i(t)^{\mathsf{T}} + \sum_{i=1}^{R} \mathbf{q}_i(t)\Big(F_{\mathbf{x}}\mathbf{q}_i(t)\Big)^{\mathsf{T}}$$

In the first equality, $\ell_{\mathbf{xx}}$ is equal to 0, from Assumption 3.1. Additionally, taking the time derivative of $Q_{\mathbf{xx}}$ in Eq. 24 yields

$$-\frac{dQ_{\mathbf{xx}}}{dt} = -\frac{d}{dt}\Big(\sum_{i=1}^{R} \mathbf{q}_i(t)\mathbf{q}_i(t)^{\mathsf{T}}\Big) = -\sum_{i=1}^{R} \Big[\frac{d\mathbf{q}_i(t)}{dt}\mathbf{q}_i(t)^{\mathsf{T}} + \mathbf{q}_i(t)\frac{d\mathbf{q}_i(t)}{dt}^{\mathsf{T}}\Big] \quad (26)$$

Equating Equations 26 and 25 yields: $-\frac{d\mathbf{q}_i}{dt} = F_{\mathbf{x}}\mathbf{q}_i$. We proceed in similar fashion for $Q_{\mathbf{xu}}$, and $Q_{\mathbf{xv}}$.

$$-\frac{dQ_{\mathbf{xu}}}{dt} = \ell_{\mathbf{xu}} + F_{\mathbf{x}}Q_{\mathbf{xu}} + Q_{\mathbf{xx}}F_{\mathbf{u}}^{\mathsf{T}} = F_{\mathbf{x}}Q_{\mathbf{xu}} + Q_{\mathbf{xx}}F_{\mathbf{u}}^{\mathsf{T}}$$

$$= F_{\mathbf{x}}\Big(\sum_{i=1}^{R} \mathbf{q}_i(t)\mathbf{p}_i(t)^{\mathsf{T}}\Big) + \Big(\sum_{i=1}^{R} \mathbf{q}_i(t)\mathbf{q}_i(t)^{\mathsf{T}}\Big)F_{\mathbf{u}}^{\mathsf{T}} \quad (27)$$

$$= \sum_{i=1}^{R} \Big(F_{\mathbf{x}}\mathbf{q}_i(t)\Big)\mathbf{p}_i(t)^{\mathsf{T}} + \sum_{i=1}^{R} \mathbf{q}_i(t)\Big(F_{\mathbf{u}}\mathbf{q}_i(t)\Big)^{\mathsf{T}}$$

$$-\frac{dQ_{\mathbf{xv}}}{dt} = \ell_{\mathbf{xv}} + F_{\mathbf{x}}Q_{\mathbf{xv}} + Q_{\mathbf{xx}}F_{\mathbf{v}}^{\mathsf{T}} = F_{\mathbf{x}}Q_{\mathbf{xv}} + Q_{\mathbf{xx}}F_{\mathbf{v}}^{\mathsf{T}}$$

$$= F_{\mathbf{x}}\Big(\sum_{i=1}^{R} \mathbf{q}_i(t)\mathbf{s}_i(t)^{\mathsf{T}}\Big) + \Big(\sum_{i=1}^{R} \mathbf{q}_i(t)\mathbf{q}_i(t)^{\mathsf{T}}\Big)F_{\mathbf{v}}^{\mathsf{T}} \quad (28)$$

$$= \sum_{i=1}^{R} \Big(F_{\mathbf{x}}\mathbf{q}_i(t)\Big)\mathbf{s}_i(t)^{\mathsf{T}} + \sum_{i=1}^{R} \mathbf{q}_i(t)\Big(F_{\mathbf{v}}\mathbf{q}_i(t)\Big)^{\mathsf{T}}$$

In similar fashion, we also compute the time derivative for $Q_{\mathbf{xu}}$ and $Q_{\mathbf{xv}}$.

$$-\frac{dQ_{\mathbf{xu}}}{dt} = -\frac{d}{dt}\Big(\sum_{i=1}^{R}\mathbf{q}_i(t)\mathbf{p}_i(t)^{\mathsf{T}}\Big) = -\sum_{i=1}^{R}\Big[\frac{d\mathbf{q}_i(t)}{dt}\mathbf{p}_i(t)^{\mathsf{T}} + \mathbf{q}_i(t)\frac{d\mathbf{p}_i(t)}{dt}^{\mathsf{T}}\Big] \tag{29}$$

$$-\frac{dQ_{\mathbf{xv}}}{dt} = -\frac{d}{dt}\Big(\sum_{i=1}^{R}\mathbf{q}_i(t)\mathbf{s}_i(t)^{\mathsf{T}}\Big) = -\sum_{i=1}^{R}\Big[\frac{d\mathbf{q}_i(t)}{dt}\mathbf{s}_i(t)^{\mathsf{T}} + \mathbf{q}_i(t)\frac{d\mathbf{s}_i(t)}{dt}^{\mathsf{T}}\Big] \tag{30}$$

It follows that

$$-\frac{d\mathbf{p}_i}{dt} = F_{\mathbf{u}}\mathbf{q}_i, \text{ and } -\frac{d\mathbf{s}_i}{dt} = F_{\mathbf{v}}\mathbf{q}_i \tag{31}$$

$\square$

Based on the results above, we can deduce that the Hessian $Q_{\mathbf{uu}}$ can be rewritten as follows

$$-\frac{dQ_{\mathbf{uu}}}{dt} = \ell_{\mathbf{uu}} + F_{\mathbf{u}}Q_{\mathbf{xu}} + Q_{\mathbf{ux}}F_{\mathbf{u}}^{\mathsf{T}}$$

$$= R_{\mathbf{u}}I + \sum_{i=1}^{R}\Big(F_{\mathbf{u}}\mathbf{q}_i(t)\Big)\mathbf{p}_i(t)^{\mathsf{T}} + \sum_{i=1}^{R}\mathbf{p}_i(t)\Big(F_{\mathbf{u}}\mathbf{q}_i(t)\Big)^{\mathsf{T}} \tag{32}$$

Integrating the equation above for $t \in [t, t_f]$ yields

$$\int_{t}^{t_f} -\frac{dQ_{\mathbf{uu}}}{dt}dt = \int_{t}^{t_f} R_{\mathbf{u}}Idt + \int_{t}^{t_f}\Big(\sum_{i=1}^{R}\Big(F_{\mathbf{u}}\mathbf{q}_i(t)\Big)\mathbf{p}_i(t)^{\mathsf{T}} + \sum_{i=1}^{R}\mathbf{p}_i(t)\Big(F_{\mathbf{u}}\mathbf{q}_i(t)\Big)^{\mathsf{T}}\Big) \tag{33}$$

Recall the terminal condition $Q_{\mathbf{uu}}(t_f) = 0$, and using Eq. 31, we obtain

$$Q_{\mathbf{uu}}(t) = R_{\mathbf{u}}I(t_f - t) - \sum_{i=1}^{R}\int_{t}^{t_f}\frac{d\mathbf{p}_i}{dt}\mathbf{p}_i^{\mathsf{T}} + \mathbf{p}_i\Big(\frac{d\mathbf{p}_i}{dt}\Big)^{\mathsf{T}}dt$$

$$= R_{\mathbf{u}}I(t_f - t) + \sum_{i=1}^{R}\int_{t_f}^{t}\frac{d}{dt}(\mathbf{p}_i\mathbf{p}_i^{\mathsf{T}})dt \tag{34}$$

$$= R_{\mathbf{u}}I(t_f - t) + \sum_{i=1}^{R}\mathbf{p}_i(t)\mathbf{p}_i^{\mathsf{T}}(t)$$

Therefore second order matrices can be rewritten as follows:

$$Q_{\mathbf{uu}}(t) = R_{\mathbf{u}}I(t_f - t) + \sum_{i=1}^{R}\Big(\int_{t_f}^{t}F_{\mathbf{u}}\mathbf{q}_idt\Big)\Big(\int_{t_f}^{t}F_{\mathbf{u}}\mathbf{q}_idt\Big)^{\mathsf{T}} \tag{35a}$$

$$Q_{\mathbf{vv}}(t) = R_{\mathbf{v}}I(t_f - t) + \sum_{i=1}^{R}\Big(\int_{t_f}^{t}F_{\mathbf{v}}\mathbf{q}_idt\Big)\Big(\int_{t_f}^{t}F_{\mathbf{v}}\mathbf{q}_idt\Big)^{\mathsf{T}} \tag{35b}$$

$$Q_{\mathbf{uv}}(t) = \sum_{i=1}^{R}\Big(\int_{t_f}^{t}F_{\mathbf{u}}\mathbf{q}_idt\Big)\Big(\int_{t_f}^{t}F_{\mathbf{v}}\mathbf{q}_idt\Big)^{\mathsf{T}} \tag{35c}$$

$$Q_{\mathbf{ux}}(t) = \sum_{i=1}^{R}\Big(\int_{t_f}^{t}F_{\mathbf{x}}\mathbf{q}_idt\Big)\Big(\int_{t_f}^{t}F_{\mathbf{u}}\mathbf{q}_idt\Big)^{\mathsf{T}}, Q_{\mathbf{vx}}(t) = \sum_{i=1}^{R}\Big(\int_{t_f}^{t}F_{\mathbf{x}}\mathbf{q}_idt\Big)\Big(\int_{t_f}^{t}F_{\mathbf{v}}\mathbf{q}_idt\Big)^{\mathsf{T}} \tag{35d}$$

## A.3 LAYER WISE PARTITIONING

To avoid dimensionality issues and tensor representations, the integrations in Eq. (35) are separated into each layer $j$ of the network. We denote $\mathbf{a}^j$, $\mathbf{h}^j$, $\mathbf{u}^j$, and $\mathbf{v}^j$ as the activation, pre-activation (linear combination), and the parameters of layer $j$, respectively. Furthermore, we consider the the preactivation vector $\mathbf{h}_j(t)$, as an affine combination of the weights with the input to the $j^{\text{th}}$ layer. We recall that we have symmetric layer-wise dynamics with respect to $\mathbf{u}$, and $\mathbf{v}$, resulting in the

partial derivatives of the preactivation vector at the $j^{\text{th}}$ layer with respect to the control variables: $\mathbf{h}_{\mathbf{u}}^j = \mathbf{h}_{\mathbf{v}}^j = \mathbf{a}_j$, and $\mathbf{h}_{\mathbf{x}}^j = (\mathbf{u} + \mathbf{v})$. This implies that:

$$F_{\mathbf{x}_j}\mathbf{q}_i = (\mathbf{u} + \mathbf{v})^{\mathsf{T}}(\frac{\partial F}{\partial \mathbf{h}_j}\mathbf{q}_i) \text{ for Feed Forward layers} \tag{36a}$$

$$F_{\mathbf{u}_j}\mathbf{q}_i == \frac{\partial F}{\partial \mathbf{h}_j}(\mathbf{a}_j \otimes I)\mathbf{q}_i = \mathbf{a}_j \otimes (\frac{\partial F}{\partial \mathbf{h}_j}\mathbf{q}_i) \text{ for Feed Forward layers} \tag{36b}$$

$$F_{\mathbf{x}_j}\mathbf{q}_i = (\mathbf{u} + \mathbf{v})\hat{*}(\frac{\partial F}{\partial \mathbf{h}_j}\mathbf{q}_i) \text{ for the Convolution layers} \tag{36c}$$

$$F_{\mathbf{u}_j}\mathbf{q}_i = \mathbf{a}_j\hat{*}(\frac{\partial F_{u_j}}{\partial \mathbf{h}_j}\mathbf{q}_i) \text{ for the Convolution layers} \tag{36d}$$

where $\otimes$ denotes the Kronecker product, and $\hat{*}$ denotes the de-convolution operator. The layer-wise notation used above was adopted as a manner to circumvent dimensionality and tensor representations issues. In this vein, the integrations in equations (35) are broken down into each layer $j$ of the network structure, where using the expression from 36, we obtain

$$\int_{t_f}^{t_0} \left(F_{\mathbf{x}}\mathbf{q}_i\right)dt = \left[\ldots, \int_{t_f}^{t_0} \left(F_{\mathbf{x}^n}\mathbf{q}_i\right)dt, \ldots\right] = \left[\ldots, \int_{t_f}^{t_0} \left((\mathbf{u} + \mathbf{v})^{\mathsf{T}}(\frac{\partial F}{\partial \mathbf{h}_j}\mathbf{q}_i)\right)dt, \ldots\right] \tag{37a}$$

$$\int_{t_f}^{t_0} \left(F_{\mathbf{u}}\mathbf{q}_i\right)dt = \left[\ldots, \int_{t_f}^{t_0} \left(F_{u^n}\mathbf{q}_i\right)dt, \ldots\right] = \left[\ldots, \int_{t_f}^{t_0} \left(\mathbf{a}_j \otimes (\frac{\partial F}{\partial \mathbf{h}_j}\mathbf{q}_i)\right)dt, \ldots\right] \tag{37b}$$

$$\int_{t_f}^{t_0} \left(F_{\mathbf{v}}\mathbf{q}_i\right)dt = \left[\ldots, \int_{t_f}^{t_0} \left(F_{\mathbf{v}^n}\mathbf{q}_i\right)dt, \ldots\right] = \left[\ldots, \int_{t_f}^{t_0} \left(\mathbf{a}_j \otimes (\frac{\partial F}{\partial \mathbf{h}_j}\mathbf{q}_i)\right)dt, \ldots\right] \tag{37c}$$

### A.4 DERIVATION OF EQUATION 10

Following the layer-wise representation, We recall that to derive 10 these expressions, we first use the Kronecker product property: $(A \otimes B)(C \otimes D) = AC \otimes BD$. Additionally, during the process of this derivation some approximations are necessary. The following assumptions are necessary in order to derive

**Assumption A.1.** Suppose:

- $\mathbf{a}(t)_j$, and $\mathbf{g}(t)_j$ are uncorrelated across time
- $\mathbf{a}_j(t)$, $\mathbf{g}_j(t)$ are pair-wise independent (Liu et al. 2021 NIPS)

For the sake of brevity we demonstrate the analytic proof only for the Hessian $Q_{\mathbf{uu}}$, the others follow similarly.

$$\begin{aligned}
Q_{\mathbf{u}_j\mathbf{u}_j}(t_0) &= R_{\mathbf{u}}I(t_0 - t_f) + \sum_{i=1}^{R}\Big(\int_{t_f}^{t} F_{\mathbf{u}}\mathbf{q}_i dt\Big)\Big(\int_{t_f}^{t} F_{\mathbf{u}}\mathbf{q}_i dt\Big)^{\mathsf{T}} \\
&= R_{\mathbf{u}}I(t_0 - t_f) + \sum_{i=1}^{R}\Big(\int_{t_f}^{t} \mathbf{a}_j \otimes (\frac{\partial F}{\partial \mathbf{h}_j}\mathbf{q}_i)dt\Big)\Big(\int_{t_f}^{t} \mathbf{a}_j \otimes (\frac{\partial F}{\partial \mathbf{h}_j}\mathbf{q}_i)dt\Big)^{\mathsf{T}} \\
&\approx R_{\mathbf{u}}I(t_0 - t_f) + \sum_{i=1}^{R}\int_{t_f}^{t} \Big(\mathbf{a}_j \otimes (\frac{\partial F}{\partial \mathbf{h}_j}\mathbf{q}_i)\Big)\Big(\mathbf{a}_j \otimes (\frac{\partial F}{\partial \mathbf{h}_j}\mathbf{q}_i)dt\Big)^{\mathsf{T}} \\
&= R_{\mathbf{u}}I(t_0 - t_f) + \sum_{i=1}^{R}\int_{t_f}^{t} \Big(\mathbf{a}_j\mathbf{a}_j^{\mathsf{T}}\otimes(\frac{\partial F}{\partial \mathbf{h}_j}\mathbf{q}_i)(\frac{\partial F}{\partial \mathbf{h}_j}\mathbf{q}_i)^{\mathsf{T}}\Big)dt \\
&\approx R_{\mathbf{u}}I(t_0 - t_f) + \underbrace{\int_{t_f}^{t_0} \Big(\mathbf{a}_j\mathbf{a}_j^{\mathsf{T}}\Big)dt}_{A_j(t)} \otimes \underbrace{\int_{t_f}^{t_0} \sum_{i=1}^{R}\Big((\frac{\partial F}{\partial \mathbf{h}_j}\mathbf{q}_i)(\frac{\partial F}{\partial \mathbf{h}_j}\mathbf{q}_i)^{\mathsf{T}}\Big)}_{B_j(t)}
\end{aligned} \tag{38}$$

*Remark* A.2. We begin by underlining that both assumptions are widely adopted (Martens & Grosse (2015)). The first assumption admits that $\mathbf{a}_j(t) \otimes \mathbf{g}_j(t)$ are temporally uncorrelated. This is necessary to yield tractable Kronecker matrices for second-order optimization. Although, it may be a strong assumption, in some cases, it has been empirically observed that the uncorrelated temporal assumption may yield better performance Laurent et al. (2018)). The second assumption admits that $\mathbf{a}_j(t)$ and $\mathbf{g}_j(t)$ are pair-wise independents, which has been verified empirically in Wu et al. (2020).

## A.5   PROOF OF PROPOSITION 3.6

*Proof.* We begin by trying to simplifying the expressions for $Q_{\mathbf{uu}}$, and $Q_{\mathbf{vv}}$. Setting: $(\Sigma_A \otimes \Sigma_B) = S_{uv}$, $(\Sigma_A \otimes \Sigma_B + \lambda_u) = S_u$, and $(\Sigma_A \otimes \Sigma_B + \lambda_v) = S_v$, we obtain:

$$
\begin{aligned}
\tilde{Q}_{\mathbf{uu}} &= (Q_{\mathbf{uu}} - Q_{\mathbf{uv}} Q_{\mathbf{vv}}^{-1} Q_{\mathbf{vu}}) \\
&= (U_A \otimes U_B) S_u (U_A \otimes U_B)^\intercal - (U_A \otimes U_B) S_{uv} (U_A \otimes U_B)^\intercal \\
&\quad (U_A \otimes U_B) S_v^{-1} (U_A \otimes U_B)^\intercal (U_A \otimes U_B) S_{uv} (U_A \otimes U_B)^\intercal \\
&= (U_A \otimes U_B) S_u (U_A \otimes U_B)^\intercal - (U_A \otimes U_B)(S_{uv} S_v^{-1} S_{uv})(U_A \otimes U_B)^\intercal \\
&= (U_A \otimes U_B)(S_u - S_{uv} S_v^{-1} S_{uv})(U_A \otimes U_B)^\intercal
\end{aligned}
\tag{39}
$$

Similarly, for $\tilde{Q}_{\mathbf{vv}} = (U_A \otimes U_B)(S_v - S_{uv} S_u^{-1} S_{uv})(U_A \otimes U_B)^\intercal$. At this point, using the following properties of the Kronecker product, based on the eigenvalue decomposition:

$$
(A \otimes B + \lambda I) = (U_A \otimes U_B)(\Sigma_A \otimes \Sigma_B + \lambda)(U_A \otimes U_B)^\intercal \tag{40}
$$

$$
(A \otimes B + \lambda I)^{-1} = (U_A \otimes U_B)(\Sigma_A \otimes \Sigma_B + \lambda)^{-1}(U_A \otimes U_B)^\intercal \tag{41}
$$

we can easily obtain the inverse of the matrices in 39 as follows:

$$
\begin{aligned}
\tilde{Q}_{\mathbf{uu}}^{-1} &= (U_A \otimes U_B)(S_u - S_{uv} S_v^{-1} S_{uv})^{-1}(U_A \otimes U_B)^\intercal, \\
\tilde{Q}_{\mathbf{vv}}^{-1} &= (U_A \otimes U_B)(S_v - S_{uv} S_u^{-1} S_{uv})^{-1}(U_A \otimes U_B)^\intercal.
\end{aligned}
$$

$\square$

## A.6   PROOF OF PROPOSITION 3.7

*Proof.* We begin by considerin the closed loop Min-Max DDP update law

$$
\begin{bmatrix} \delta\mathbf{u}_t \\ \delta\mathbf{v}_t \end{bmatrix} = \begin{bmatrix} l_{\mathbf{u}} \\ l_{\mathbf{v}} \end{bmatrix} + \begin{bmatrix} K_{\mathbf{u}} \\ K_{\mathbf{v}} \end{bmatrix} \delta\mathbf{x}_t,
\tag{42}
$$

where the feed-forward gains $l_{\mathbf{u}}$, along with $l_{\mathbf{v}}$, and the feedback gains $K_{\mathbf{u}}$, $K_{\mathbf{v}}$ are given by

$$
\begin{aligned}
l_{\mathbf{u}} &= \tilde{Q}_{\mathbf{uu}}^{-1}(Q_{\mathbf{uv}} Q_{\mathbf{vv}}^{-1} Q_{\mathbf{v}} - Q_{\mathbf{u}}), \text{ and } K_{\mathbf{u}} = \tilde{Q}_{\mathbf{uu}}^{-1}(Q_{\mathbf{ux}} - Q_{\mathbf{uv}} Q_{\mathbf{vv}}^{-1} Q_{\mathbf{vx}}) \\
l_{\mathbf{v}} &= \tilde{Q}_{\mathbf{vv}}^{-1}(Q_{\mathbf{uv}} Q_{\mathbf{uu}}^{-1} Q_{\mathbf{u}} - Q_{\mathbf{v}}), \text{ and } K_{\mathbf{v}} = \tilde{Q}_{\mathbf{vv}}^{-1}(Q_{\mathbf{vx}} - Q_{\mathbf{uv}} Q_{\mathbf{vv}}^{-1} Q_{\mathbf{vx}})
\end{aligned}
\tag{43}
$$

with $\tilde{Q}_{\mathbf{uu}} = (Q_{\mathbf{uu}} - Q_{\mathbf{uv}} Q_{\mathbf{vv}}^{-1} Q_{\mathbf{vu}})$, and $\tilde{Q}_{\mathbf{vv}} = (Q_{\mathbf{vv}} - Q_{\mathbf{vu}} Q_{\mathbf{uu}}^{-1} Q_{\mathbf{uv}})$.

At this point, we will leverage the following property of the Kronecker products: $(A \otimes B)\mathbf{r} = \text{vec}(BRA^\intercal)$, where $R = \text{vec}^{-1}(r)$ denotes the inverse vectorization of $\mathbf{r}$, to express $\ell_{\mathbf{u}}$, and $\ell_{\mathbf{v}}$ in a compact manner.

Setting for brevity: $Q_{\mathbf{ux}}\delta\mathbf{x}_t = \delta\mathbf{q_u}$, $Q_{\mathbf{vx}}\delta\mathbf{x}_t = \delta\mathbf{q_v}$, and substituting into 42 the expression for 13, the feed-forward and feedback gains in (43) can be compactly rewritten as follows

$$l_{\mathbf{u}} = (U_A \otimes U_B) \underbrace{(S_u - S_{uv}S_v^{-1}S_{uv})^{-1}(S_{uv}S_v^{-1})\text{vec}(U_B^\mathsf{T}\bar{Q}_{\mathbf{v}}U_A)}_{\mathbf{g}_{uv}}$$
$$- (U_A \otimes U_B) \underbrace{(S_u - S_{uv}S_v^{-1}S_{uv})^{-1}\text{vec}(U_B^\mathsf{T}\bar{Q}_{\mathbf{u}}U_A)}_{\mathbf{g}_{uu}} = \text{vec}(U_B G_{uv}U_A^\mathsf{T}) - \text{vec}(U_B G_{uu}U_A^\mathsf{T})$$

(44a)

$$l_{\mathbf{v}} = (U_A \otimes U_B) \underbrace{(S_v - S_{uv}S_u^{-1}S_{uv})^{-1}(S_{uv}S_u^{-1})\text{vec}(U_B^\mathsf{T}\bar{Q}_{\mathbf{u}}U_A)}_{\mathbf{g}_{vu}}$$
$$- (U_A \otimes U_B) \underbrace{(S_u - S_{uv}S_v^{-1}S_{uv})^{-1}\text{vec}(U_B^\mathsf{T}\bar{Q}_{\mathbf{v}}U_A)}_{\mathbf{g}_{vv}} = \text{vec}(U_B G_{vu}U_A^\mathsf{T}) - \text{vec}(U_B G_{vv}U_A^\mathsf{T})$$

(44b)

$$K_{\mathbf{u}} = (U_A \otimes U_B) \underbrace{(S_u - S_{uv}S_v^{-1}S_{uv})^{-1}(S_{uv}S_v^{-1})\delta\mathbf{q}_{\mathbf{v}}}_{\mathbf{y}_{vv}}$$
$$- (U_A \otimes U_B) \underbrace{(S_u - S_{uv}S_v^{-1}S_{uv})^{-1}\delta\mathbf{q}_{\mathbf{u}}}_{\mathbf{y}_{vv}} = \text{vec}(U_B Y_{uv}U_A^\mathsf{T}) - \text{vec}(U_B Y_{uu}U_A^\mathsf{T})$$

(44c)

$$K_{\mathbf{v}} = (U_A \otimes U_B) \underbrace{(S_u - S_{uv}S_v^{-1}S_{uv})^{-1}(S_{uv}S_v^{-1})\delta\mathbf{q}_{\mathbf{u}}}_{\mathbf{y}_{vv}}$$
$$- (U_A \otimes U_B) \underbrace{(S_u - S_{uv}S_v^{-1}S_{uv})^{-1}\delta\mathbf{q}_{\mathbf{v}}}_{\mathbf{y}_{vv}} = \text{vec}(U_B Y_{vu}U_A^\mathsf{T}) - \text{vec}(U_B Y_{vv}U_A^\mathsf{T})$$

(44d)

where $\text{vec}^{-1}(Q_i) = \bar{Q}_i$, $\text{vec}^{-1}(g_{ij}) = G_{ij}$, and $\text{vec}^{-1}(\mathbf{y}_{ij}) = Y_{ij}$ for $i = \{\mathbf{u}, \mathbf{v}\}$ denote the inverse of the vectorization operation of the corresponding vectors. The only required inversions are of the diagonal S matrices which are computationally inexpensive, rendering the computation of the terms in 44 tractable to compute. $\qquad\square$

# B  DIFFERENTIAL DYNAMIC PROGRAMMING IN CONTINUOUS TIME

## B.1  PROBLEM FORMULATION

Consider a min-max game problem, with dynamics described the following ODE:

$$\frac{d\mathbf{x}(t)}{dt} = F(\mathbf{x}(t), \mathbf{u}(t), \mathbf{v}(t), t), \quad x(t_0) = x_0 \tag{45}$$

where $x(t) \in \mathcal{X}$ is the state of the dynamic system at $t \in [t_0, t_f]$, and $\mathbf{u}(t, \mathbf{u}(t)) \in U_1 \subset \mathcal{U}$ and $\mathbf{v}(t, \mathbf{x}(t)) \in U_2 \subset V$ denote conflicting controls, with $U_1$ and $U_2$ are convex sets containing all admissible controls of $\mathbf{u}$ and $\mathbf{v}$ respectively. For the sake of brevity, we will denote $\mathbf{u}(t, \mathbf{x}(t)) \equiv \mathbf{u}$, and similarly $\mathbf{v}(t, \mathbf{x}(t)) = \mathbf{v}$. The goal of this OCP scheme is to find non-anticipating strategies for both players. Additionally, we define the cost function as follows:

$$J(\mathbf{u}, \mathbf{v}) = \phi(t_f, \mathbf{x}_{t_f}) + \int_{t_0}^{t_f} \ell(\mathbf{x}, \mathbf{u}, \mathbf{v}, t)d\tau \tag{46}$$

where $\phi : [t_0, t_f] \times \mathcal{X} \to \mathbb{R}_+$ is the terminal cost, and $\ell : \mathcal{X} \times \mathcal{U} \times V \times [t_0, t_f] \to \mathbb{R}_+$ is the running cost incorporating the state and the control cost for both players. The conflict between the two players enters in our problem formulation as one tries through control $\mathbf{u}$ to minimize the above cost function, whereas the other one tries to maximize it through control $\mathbf{v}$. We proceed to define the value function for our problem as the function expressing the minmax value of the cost function at time $t = t_0$ and $x = x_0$.

$$V(t_0, x_0) = \min_{\mathbf{u}} \max_{\mathbf{v}} \left\{ \phi(t_f, \mathbf{x}_{t_f}) + \int_{t_0}^{t_f} \ell(\mathbf{x}, \mathbf{u}, \mathbf{v}, t)d\tau \right\} \tag{47}$$

Using the Bellman principle, we write (47), as follows:

$$
\begin{aligned}
V(t, x(t)) &= \min_{\mathbf{u}(t \to t_f)} \max_{\mathbf{v}(t \to t_f)} \left\{ \phi(t_f, \mathbf{x}_{t_f}) + \int_t^{t+dt} \ell(\mathbf{x}, \mathbf{u}, \mathbf{v}, t) d\tau + \int_{t+dt}^{t_f} \ell(\mathbf{x}, \mathbf{u}, \mathbf{v}, t) d\tau \right\} \\
&= \min_{\mathbf{u}(t \to t+dt)} \max_{\mathbf{v}(t \to t+dt)} \left\{ \min_{\mathbf{u}(t+dt \to t_f)} \max_{\mathbf{v}(t+dt \to t_f)} \left\{ \phi(t_f, \mathbf{x}_{t_f}) + \int_{t+dt}^{t_f} \ell(\mathbf{x}, \mathbf{u}, \mathbf{v}, t) d\tau \right\} \right. \\
&\quad \left. + \int_t^{t+dt} \ell(\mathbf{x}, \mathbf{u}, \mathbf{v}, t) d\tau \right\} \\
&= \min_{\mathbf{u}(t \to t+dt)} \max_{\mathbf{v}(t \to t+dt)} \left\{ \int_t^{t+dt} \ell(\mathbf{x}, \mathbf{u}, \mathbf{v}, t) d\tau + V(t+dt, \mathbf{x}_{t+dt}) \right\}
\end{aligned}
\tag{48}
$$

Then we can readily obtain:

$$
0 = \min_{\mathbf{u}} \max_{\mathbf{v}} \left[ \ell(\mathbf{x}, \mathbf{u}, \mathbf{v}, t) + \frac{dV}{dt} \right]
\tag{49}
$$

We can express $dV$ as follows:

$$
dV \approx \frac{\partial V}{\partial t} dt + \frac{\partial V}{\partial \mathbf{x}_t}^{\mathsf{T}} d\mathbf{x}_t = \frac{\partial V}{\partial t} dt + \frac{\partial V}{\partial \mathbf{x}_t}^{\mathsf{T}} F dt
\tag{50}
$$

This form enables us to obtain the min-max Hamilton Jacobi Bellman through substitution to (49):

$$
\begin{aligned}
0 &= \min_{\mathbf{u}} \max_{\mathbf{v}} \left[ \ell(\mathbf{x}, \mathbf{u}, \mathbf{v}, t) + \frac{\partial V}{\partial t} + \frac{\partial V}{\partial \mathbf{x}_t}^{\mathsf{T}} F \right] => \\
-\frac{\partial V}{\partial t} &= \min_{\mathbf{u}} \max_{\mathbf{v}} \left[ \ell(\mathbf{x}(t), \mathbf{u}, \mathbf{v}, t) + \frac{\partial V}{\partial \mathbf{x}_t}^{\mathsf{T}} F \right]
\end{aligned}
\tag{51}
$$

with its terminal condition being: $V(t_f, \mathbf{x}_{t_f}) = \phi(t_f, \mathbf{x}_{t_f})$.

## B.2 Backwards Propagation

First, we consider equation function $Q$ as: $Q(\mathbf{x}, \mathbf{u}, \mathbf{v}, t) = \ell(\mathbf{x}, \mathbf{u}, \mathbf{v}, t) + \frac{dV}{dt}$. We expand the terms inside the minimization in (49) along with function Q up to second order terms, with respect to the nomimal trajectory $(\bar{\mathbf{x}}, \bar{\mathbf{u}}, \bar{\mathbf{v}})$ and we obtain:

$$
\ell(\bar{\mathbf{x}} + \delta\mathbf{x}_t, \bar{\mathbf{u}} + \delta\mathbf{u}_t, \bar{\mathbf{v}} + \delta\mathbf{v}_t) =
$$
$$
\ell(\bar{\mathbf{x}}, \bar{\mathbf{u}}, \bar{\mathbf{v}}, t) + \ell_{\mathbf{x}}^{\mathsf{T}} \delta\mathbf{x} + \ell_{\mathbf{u}}^{\mathsf{T}} \delta\mathbf{u} + \ell_{\mathbf{v}}^{\mathsf{T}} \delta\mathbf{v}_t + \frac{1}{2} \begin{bmatrix} \delta\mathbf{x}_t \\ \delta\mathbf{u}_t \\ \delta\mathbf{v}_t \end{bmatrix}^{\mathsf{T}} \begin{bmatrix} \ell_{\mathbf{xx}} & \ell_{\mathbf{xu}} & \ell_{\mathbf{xv}} \\ \ell_{\mathbf{ux}} & \ell_{\mathbf{uu}} & \ell_{\mathbf{uv}} \\ \ell_{\mathbf{vx}} & \ell_{\mathbf{vu}} & \ell_{\mathbf{vv}} \end{bmatrix} \begin{bmatrix} \delta\mathbf{x}_t \\ \delta\mathbf{u}_t \\ \delta\mathbf{v}_t \end{bmatrix}
\tag{52}
$$

$$
V(t+dt, \bar{\mathbf{x}} + \delta\mathbf{x}_t) = V(t, \bar{\mathbf{x}}) + V_{\mathbf{x}}(t, \mathbf{x}) \delta\mathbf{x}_t + \frac{1}{2} \delta\mathbf{x}_t^{\mathsf{T}} V_{\mathbf{xx}} \delta\mathbf{x}_t
\tag{53}
$$

At this point, we take the derivative with respect to time, of the expanded expression of the value function from (23). We note that $V, V_{\mathbf{x}}, V_{\mathbf{xx}}$ are only functions of time, since they are expanded with respect to the nominal $\bar{\mathbf{x}}(t)$. Additionally, we also notice that from the system dynamics the infinitesimal perturbation around the nominal trajectory $\delta\mathbf{x}_t$ are also a function of time, so the differentiation of the second and third term below follow the product differentiation rule.

$$
\begin{aligned}
\frac{dV}{dt} &= \frac{dV}{dt}\Big|_{\substack{x=\bar{\mathbf{x}}(t) \\ u=\bar{\mathbf{u}}(t) \\ v=\bar{\mathbf{v}}(t)}} + \frac{d}{dt}\left( V_{\mathbf{x}} \delta\mathbf{x}_t \right) + \frac{d}{dt}\left( \frac{1}{2} \delta\mathbf{x}^{\mathsf{T}} V_{\mathbf{xx}} \delta\mathbf{x}_t \right) \\
&= \frac{dV}{dt} + \left( \frac{dV}{dt}^{\mathsf{T}} \delta\mathbf{x}_t + V_{\mathbf{x}}^{\mathsf{T}} \frac{d\delta\mathbf{x}_t}{dt} \right) + \frac{1}{2}\left( \frac{d\delta\mathbf{x}}{dt}^{\mathsf{T}} V_{\mathbf{xx}} \delta\mathbf{x}_t + \delta\mathbf{x}_t^{\mathsf{T}} \frac{dV_{\mathbf{xx}}}{dt} \delta\mathbf{x}_t + \delta\mathbf{x}_t^{\mathsf{T}} V_{\mathbf{xx}} \frac{d\delta\mathbf{x}_t}{dt} \right)
\end{aligned}
\tag{54}
$$

Returning to the system dynamics, we can write $\frac{d\delta\mathbf{x}}{dt}$, as follows:

$$
\begin{aligned}
\frac{d\delta\mathbf{x}}{dt} &= \frac{d}{dt}\Big(\mathbf{x}(t) - \bar{\mathbf{x}}(t)\Big) = F(\mathbf{x}, \mathbf{u}, \mathbf{v}, t) - F(\bar{\mathbf{x}}, \bar{\mathbf{u}}, \bar{\mathbf{v}}, t) \\
&= F(\bar{\mathbf{x}}, \bar{\mathbf{u}}, \bar{\mathbf{v}}, t) + \bar{F}_{\mathbf{x}}\delta\mathbf{x}_t + \bar{F}_{\mathbf{u}}\delta\mathbf{u}_t + \bar{F}_{\mathbf{v}}\delta\mathbf{v}_t - F(\bar{\mathbf{x}}, \bar{\mathbf{u}}, \bar{\mathbf{v}}, t) \\
&= \bar{F}_{\mathbf{x}}\delta\mathbf{x}_t + \bar{F}_{\mathbf{u}}\delta\mathbf{u}_t + \bar{F}_{\mathbf{v}}\delta\mathbf{v}_t
\end{aligned}
\tag{55}
$$

where $\bar{F}_{\mathbf{x}}, \bar{F}_{\mathbf{u}}, \bar{F}_{\mathbf{v}}$ stands for $F_{\mathbf{x}}(\bar{\mathbf{x}}, \bar{\mathbf{u}}, \bar{\mathbf{v}}), F_{\mathbf{u}}(\bar{\mathbf{x}}, \bar{\mathbf{u}}, \bar{\mathbf{v}}), F_{v}(\bar{\mathbf{x}}, \bar{\mathbf{u}}, \bar{\mathbf{v}})$ respectively. If we substitute (55) in (54), we obtain:

$$
\begin{aligned}
\frac{dV}{dt} = {} & \frac{dV}{dt}\Big|_{\substack{\mathbf{x}=\bar{\mathbf{x}}(t)\\\mathbf{u}=\bar{\mathbf{u}}(t)\\\mathbf{v}=\bar{\mathbf{v}}(t)}} + \Big(\frac{dV}{dt}^{\mathsf{T}}\delta\mathbf{x}_t + V_{\mathbf{x}}^{\mathsf{T}}(\bar{F}_{\mathbf{x}}\delta\mathbf{x}_t + \bar{F}_{\mathbf{u}}\delta\mathbf{u}_t + \bar{F}_{\mathbf{v}}\delta\mathbf{v}_t)\Big) \\
& + \frac{1}{2}\Big(((\bar{F}_{\mathbf{x}}\delta\mathbf{x}_t + \bar{F}_{\mathbf{u}}\delta\mathbf{u}_t + \bar{F}_{\mathbf{v}}\delta\mathbf{v}_t))^{\mathsf{T}}V_{\mathbf{xx}}\delta\mathbf{x}_t + \delta\mathbf{x}_t^{\mathsf{T}}\frac{dV_{\mathbf{xx}}}{dt}\delta\mathbf{x}_t + \delta\mathbf{x}_t^{\mathsf{T}}V_{\mathbf{xx}}((\bar{F}_{\mathbf{x}}\delta\mathbf{x}_t + \bar{F}_{\mathbf{u}}\delta\mathbf{u}_t + \bar{F}_{\mathbf{v}}\delta\mathbf{v}_t))\Big)
\end{aligned}
\tag{56}
$$

Now, if we substitute (56) and (52) in (49), separate the terms that differentiate with respect to time the value function or any of its derivatives with respect to the state, we obtain:

$$
\begin{aligned}
& -\frac{dV}{dt} - \frac{dV_{\mathbf{x}}}{dt}^{\mathsf{T}}\delta\mathbf{x}_t - \frac{1}{2}\delta\mathbf{x}_t^{\mathsf{T}}\frac{dV_{\mathbf{xx}}}{dt}\delta\mathbf{x}_t = \\
& = \min_{\delta\mathbf{u}}\max_{\delta\mathbf{v}}\left[\ell(\bar{\mathbf{x}}, \bar{\mathbf{u}}, \bar{\mathbf{v}}, t) + \ell_{\mathbf{x}}^{\mathsf{T}}\delta\mathbf{x} + \ell_{\mathbf{u}}^{\mathsf{T}}\delta\mathbf{u} + \ell_{\mathbf{v}}^{\mathsf{T}}\delta\mathbf{v} + \frac{1}{2}\begin{bmatrix}\delta\mathbf{x}_t\\\delta\mathbf{u}_t\\\delta\mathbf{v}_t\end{bmatrix}^{\mathsf{T}}\begin{bmatrix}\ell_{\mathbf{xx}} & \ell_{\mathbf{xu}} & \ell_{\mathbf{xv}}\\\ell_{\mathbf{ux}} & \ell_{\mathbf{uu}} & \ell_{\mathbf{uv}}\\\ell_{\mathbf{vx}} & \ell_{\mathbf{vu}} & \ell_{\mathbf{vv}}\end{bmatrix}\begin{bmatrix}\delta\mathbf{x}_t\\\delta\mathbf{u}_t\\\delta\mathbf{v}_t\end{bmatrix}\right. \\
& \left. + V_{\mathbf{x}}^{\mathsf{T}}F_{\mathbf{x}}\delta\mathbf{x}_t + V_{\mathbf{x}}^{\mathsf{T}}F_{\mathbf{u}}\delta\mathbf{u}_t + V_{\mathbf{x}}^{\mathsf{T}}F_{\mathbf{v}}\delta\mathbf{v}_t + \frac{1}{2}\begin{bmatrix}\delta\mathbf{x}_t\\\delta\mathbf{u}_t\\\delta\mathbf{v}_t\end{bmatrix}^{\mathsf{T}}\begin{bmatrix}\bar{F}_{\mathbf{x}}^{\mathsf{T}}V_{\mathbf{xx}} + V_{\mathbf{xx}}\bar{F}_{\mathbf{x}} & V_{\mathbf{xx}}\bar{F}_{\mathbf{u}} & V_{\mathbf{xx}}\bar{F}_{v}\\\bar{F}_{\mathbf{u}}^{\mathsf{T}}V_{\mathbf{xx}} & 0 & 0\\\bar{F}_{\mathbf{v}}^{\mathsf{T}}V_{\mathbf{xx}} & 0 & 0\end{bmatrix}\begin{bmatrix}\delta\mathbf{x}_t\\\delta\mathbf{u}_t\\\delta\mathbf{v}_t\end{bmatrix}\right] \\
& = \min_{\delta\mathbf{u}}\max_{\delta\mathbf{v}}\left[\ell(\bar{\mathbf{x}}, \bar{\mathbf{u}}, \bar{\mathbf{v}}) + (\ell_{\mathbf{x}}^{\mathsf{T}} + V_{\mathbf{x}}^{\mathsf{T}}F_{\mathbf{x}})\delta\mathbf{x}_t + (\ell_{\mathbf{u}}^{\mathsf{T}} + V_{\mathbf{x}}^{\mathsf{T}}F_{\mathbf{u}})\delta\mathbf{u}_t + (\ell_{\mathbf{v}}^{\mathsf{T}} + V_{\mathbf{x}}^{\mathsf{T}}F_{\mathbf{v}})\delta\mathbf{v}_t\right. \\
& \left. + \frac{1}{2}\begin{bmatrix}\delta\mathbf{x}_t\\\delta\mathbf{u}_t\\\delta\mathbf{v}_t\end{bmatrix}^{\mathsf{T}}\begin{bmatrix}\ell_{\mathbf{xx}} + \bar{F}_{\mathbf{x}}^{\mathsf{T}}V_{\mathbf{xx}} + V_{\mathbf{xx}}\bar{F}_{\mathbf{x}} & \ell_{\mathbf{xu}} + V_{\mathbf{xx}}\bar{F}_{\mathbf{u}} & V_{\mathbf{xx}}\bar{F}_{v}\\\ell_{\mathbf{ux}} + \bar{F}_{\mathbf{u}}^{\mathsf{T}}V_{\mathbf{xx}} & \ell_{\mathbf{uu}} & \ell_{\mathbf{uv}}\\\ell_{\mathbf{vx}} + \bar{F}_{\mathbf{v}}^{\mathsf{T}}V_{\mathbf{xx}} & \ell_{\mathbf{vu}} & \ell_{\mathbf{vv}}\end{bmatrix}\begin{bmatrix}\delta\mathbf{x}_t\\\delta\mathbf{u}_t\\\delta\mathbf{v}_t\end{bmatrix}\right]
\end{aligned}
\tag{57}
$$

Following the definition of Q and (49), we can expand Q up to second order terms as following:

$$
0 = \min_{\mathbf{u}}\max_{\mathbf{v}}\left[Q(\bar{\mathbf{x}}, \bar{\mathbf{u}}, \bar{\mathbf{v}}, t) + Q_{\mathbf{x}}^{\mathsf{T}}\delta\mathbf{x} + Q_{\mathbf{u}}^{\mathsf{T}}\delta\mathbf{u}_t + Q_{\mathbf{v}}^{\mathsf{T}}\delta\mathbf{v}_t + \frac{1}{2}\begin{bmatrix}\delta\mathbf{x}_t\\\delta\mathbf{u}_t\\\delta\mathbf{v}_t\end{bmatrix}^{\mathsf{T}}\begin{bmatrix}Q_{\mathbf{xx}} & Q_{\mathbf{xu}} & Q_{\mathbf{xv}}\\Q_{\mathbf{ux}} & Q_{\mathbf{uu}} & Q_{\mathbf{uv}}\\Q_{\mathbf{vx}} & Q_{\mathbf{vu}} & Q_{\mathbf{vv}}\end{bmatrix}\begin{bmatrix}\delta\mathbf{x}_t\\\delta\mathbf{u}_t\\\delta\mathbf{v}_t\end{bmatrix}\right]
\tag{58}
$$

Therefore, we can equate the terms from (57) with the terms of the quadratic expansion of (58), and yield the following:

$$
\begin{aligned}
Q_0(t) &= \ell(\bar{\mathbf{x}}(t), \bar{\mathbf{u}}(t), t) \\
Q_{\mathbf{x}}(t) &= (\ell_{\mathbf{x}}^{\mathsf{T}} + V_{\mathbf{x}}^{\mathsf{T}}F_{\mathbf{x}})^{\mathsf{T}} = \ell_{\mathbf{x}} + F_{\mathbf{x}}^{\mathsf{T}}V_{\mathbf{x}} \\
Q_{\mathbf{u}}(t) &= (\ell_{\mathbf{u}}^{\mathsf{T}} + V_{\mathbf{xx}}^{\mathsf{T}}F_{\mathbf{u}})^{\mathsf{T}} = \ell_{\mathbf{u}} + F_{\mathbf{u}}^{\mathsf{T}}V_{\mathbf{x}} \\
Q_{\mathbf{v}}(t) &= (\ell_{v}^{\mathsf{T}} + V_{\mathbf{xx}}^{\mathsf{T}}F_{v})^{\mathsf{T}} = \ell_{v} + F_{v}^{\mathsf{T}}V_{\mathbf{x}} \\
Q_{\mathbf{xx}}(t) &= \ell_{\mathbf{xx}} + V_{\mathbf{xx}}F_{\mathbf{x}} + F_{\mathbf{x}}^{\mathsf{T}}V_{\mathbf{xx}} \\
Q_{\mathbf{xu}}(t) &= \ell_{\mathbf{xu}} + V_{\mathbf{xx}}F_{\mathbf{u}} = Q_{\mathbf{ux}}^{\mathsf{T}} \\
Q_{\mathbf{xv}}(t) &= \ell_{\mathbf{xv}} + V_{\mathbf{xx}}F_{v} = Q_{\mathbf{vx}}^{\mathsf{T}} \\
Q_{\mathbf{uu}}(t) &= \ell_{\mathbf{uu}} \\
Q_{\mathbf{vv}}(t) &= \ell_{\mathbf{vv}}
\end{aligned}
\tag{59}
$$

Additionally, taking the derivative in (58) with respect to $\delta \mathbf{u}_t$ and $\delta \mathbf{v}_t$ and setting them equal to 0, yields:

$$
\begin{aligned}
\delta \mathbf{u}^* &= -Q_{\mathbf{uu}}^{-1}(Q_{\mathbf{ux}}\delta \mathbf{x}_t + Q_{\mathbf{uv}}\delta \mathbf{v}_t^* + Q_{\mathbf{u}}) \\
\delta \mathbf{v}^* &= -Q_{\mathbf{vv}}^{-1}(Q_{\mathbf{vx}}\delta \mathbf{x}_t + Q_{\mathbf{vu}}\delta \mathbf{u}_t^* + Q_v)
\end{aligned}
\tag{60}
$$

However, we notice that the optimal control of the opponent is present in the expression of the optimal control of each player, so we try to eliminate this dependency:

$$
\begin{aligned}
&\delta \mathbf{u}^* = -Q_{\mathbf{uu}}^{-1}(Q_{\mathbf{ux}}\delta \mathbf{x}_t + Q_{\mathbf{uv}}(-Q_{\mathbf{vv}}^{-1}(Q_{\mathbf{vx}}\delta \mathbf{x}_t + Q_{\mathbf{vu}}\delta \mathbf{u}_t^* + Q_v)) + Q_{\mathbf{u}}) \\
&\Rightarrow (Q_{\mathbf{uu}} - Q_{\mathbf{uv}}Q_{\mathbf{vv}}^{-1}Q_{\mathbf{vu}})\delta \mathbf{u}^* = -(Q_{\mathbf{ux}} - Q_{\mathbf{uv}}Q_{\mathbf{vv}}^{-1}Q_{\mathbf{vx}})\delta \mathbf{x}_t + Q_{\mathbf{uv}}Q_{\mathbf{vv}}^{-1}Q_v - Q_{\mathbf{u}} \\
&\Rightarrow \delta \mathbf{u}^* = (Q_{\mathbf{uu}} - Q_{\mathbf{uv}}Q_{\mathbf{vv}}^{-1}Q_{\mathbf{vu}})^{-1}(Q_{\mathbf{uv}}Q_{\mathbf{vv}}^{-1}Q_v - Q_v - (Q_{\mathbf{ux}} - Q_{\mathbf{uv}}Q_{\mathbf{vv}}^{-1}Q_{\mathbf{vx}})\delta \mathbf{x}_t) \\
&\Rightarrow \delta \mathbf{u}^* = l_{\mathbf{u}} + K_{\mathbf{u}}\delta \mathbf{x}_t
\end{aligned}
\tag{61}
$$

where $l_{\mathbf{u}} = (Q_{\mathbf{uu}} - Q_{\mathbf{uv}}Q_{\mathbf{vv}}^{-1}Q_{\mathbf{vu}})^{-1}(Q_{\mathbf{uv}}Q_{\mathbf{vv}}^{-1}Q_v - Q_{\mathbf{u}})$, and $K_{\mathbf{u}} = (Q_{\mathbf{uu}} - Q_{\mathbf{uv}}Q_{\mathbf{vv}}^{-1}Q_{\mathbf{vu}})^{-1}((Q_{\mathbf{ux}} - Q_{\mathbf{uv}}Q_{\mathbf{vv}}^{-1}Q_{\mathbf{vx}})\delta \mathbf{x}_t)$.
Similarly we can express $\delta \mathbf{v}^* = l_{\mathbf{v}} + K_{\mathbf{v}}\delta \mathbf{x}_t$, where equivalently the coefficients $l_{\mathbf{v}}$, and $K_{\mathbf{v}}$ are defined as: $l_{\mathbf{v}} = (Q_{\mathbf{vv}} - Q_{\mathbf{vu}}Q_{\mathbf{uu}}^{-1}Q_{\mathbf{uv}})^{-1}(Q_{\mathbf{uv}}Q_{\mathbf{uu}}^{-1}Q_{\mathbf{u}} - Q_v)$, $K_{\mathbf{v}} = (Q_{\mathbf{vv}} - Q_{\mathbf{vu}}Q_{\mathbf{uu}}^{-1}Q_{\mathbf{uv}})^{-1}((Q_{\mathbf{vx}} - Q_{\mathbf{uv}}Q_{\mathbf{vv}}^{-1}Q_{\mathbf{vx}}))$. From (57) and (58), the value function and its first and second order derivatives with respect to $\mathbf{x}$ are expressed through the following backward ordinary differential equations:

$$
-\frac{dV}{dt} - \frac{dV_{\mathbf{x}}}{dt}^{\mathsf{T}}\delta \mathbf{x}_t - \frac{1}{2}\delta \mathbf{x}_t^{\mathsf{T}}\frac{dV_{\mathbf{xx}}}{dt}\delta \mathbf{x}_t =
$$

$$
= \min_{\mathbf{u}}\max_{\mathbf{v}} Q(\bar{\mathbf{x}}, \bar{\mathbf{u}}, \bar{\mathbf{v}}, tsss) + Q_{\mathbf{x}}^{\mathsf{T}}\delta \mathbf{x} + Q_{\mathbf{u}}^{\mathsf{T}}\delta \mathbf{u}_t + Q_v^{\mathsf{T}}\delta \mathbf{v}_t + \frac{1}{2}\begin{bmatrix}\delta \mathbf{x}_t \\ \delta \mathbf{u}_t \\ \delta \mathbf{v}_t\end{bmatrix}^{\mathsf{T}}\begin{bmatrix}Q_{\mathbf{xx}} & Q_{\mathbf{xu}} & Q_{\mathbf{xv}} \\ Q_{\mathbf{ux}} & Q_{\mathbf{uu}} & Q_{\mathbf{uv}} \\ Q_{\mathbf{vx}} & Q_{\mathbf{vu}} & Q_{\mathbf{vv}}\end{bmatrix}\begin{bmatrix}\delta \mathbf{x}_t \\ \delta \mathbf{u}_t \\ \delta \mathbf{v}_t\end{bmatrix}
$$

$$
= Q(\bar{\mathbf{x}}, \bar{\mathbf{u}}, \bar{\mathbf{v}}, t) + Q_{\mathbf{x}}^{\mathsf{T}}\delta \mathbf{x} + Q_{\mathbf{u}}^{\mathsf{T}}(l_{\mathbf{u}} + K_{\mathbf{u}}\delta \mathbf{x}_t) + Q_v^{\mathsf{T}}(l_{\mathbf{v}} + K_{\mathbf{v}}\delta \mathbf{x}_t) +
$$

$$
+ \frac{1}{2}\begin{bmatrix}\delta \mathbf{x}_t \\ (l_{\mathbf{u}} + K_{\mathbf{u}}\delta \mathbf{x}_t) \\ (l_v + K_v\delta \mathbf{x}_t)\end{bmatrix}^{\mathsf{T}}\begin{bmatrix}Q_{\mathbf{xx}} & Q_{\mathbf{xu}} & Q_{\mathbf{xv}} \\ Q_{\mathbf{ux}} & Q_{\mathbf{uu}} & Q_{\mathbf{uv}} \\ Q_{\mathbf{vx}} & Q_{\mathbf{vu}} & Q_{\mathbf{vv}}\end{bmatrix}\begin{bmatrix}\delta \mathbf{x}_t \\ (l_{\mathbf{u}} + K_{\mathbf{u}}\delta \mathbf{x}_t) \\ (l_v + K_v\delta \mathbf{x}_t)\end{bmatrix}
\tag{62}
$$

$$
= \begin{cases}
-\frac{dV}{dt} = & \bar{\ell} + l_{\mathbf{u}}^{\mathsf{T}}Q_{\mathbf{u}} + l_v^{\mathsf{T}}Q_v + \frac{1}{2}l_{\mathbf{u}}^{\mathsf{T}}Q_{\mathbf{uu}}l_{\mathbf{u}} + \frac{1}{2}l_v^{\mathsf{T}}Q_{\mathbf{vv}}l_v + l_{\mathbf{u}}^{\mathsf{T}}Q_{\mathbf{uv}}l_v \\[2mm]
-\frac{dV_{\mathbf{x}}}{dt} = & Q_{\mathbf{x}} + K_{\mathbf{u}}^{\mathsf{T}}Q_{\mathbf{u}} + K_v^{\mathsf{T}}Q_v + Q_{\mathbf{ux}}^{\mathsf{T}}l_{\mathbf{u}} + Q_{\mathbf{vx}}^{\mathsf{T}}l_v + K_{\mathbf{u}}^{\mathsf{T}}Q_{\mathbf{uu}}l_u + \\
& + K_{\mathbf{u}}^{\mathsf{T}}Q_{\mathbf{uv}}lv + K_v^{\mathsf{T}}Q_{\mathbf{vu}}l_{\mathbf{u}} + K_v^{\mathsf{T}}Q_{\mathbf{vv}}l_v \\[2mm]
-\frac{dV_{\mathbf{xx}}}{dt} = & Q_{\mathbf{xx}} + K_{\mathbf{u}}^{\mathsf{T}}Q_{\mathbf{ux}} + Q_{\mathbf{ux}}^{\mathsf{T}}K_{\mathbf{u}} + K_v^{\mathsf{T}}Q_{\mathbf{vx}} + Q_{\mathbf{vx}}^{\mathsf{T}}K_v + K_v^{\mathsf{T}}Q_{\mathbf{vu}}K_{\mathbf{u}} + \\
& + K_{\mathbf{u}}^{\mathsf{T}}Q_{\mathbf{uv}}K_v + K_{\mathbf{u}}^{\mathsf{T}}Q_{\mathbf{uu}}K_{\mathbf{u}} + K_v Q_{\mathbf{vv}}K_v
\end{cases}
\tag{63}
$$

under terminal conditions:

$$
\begin{aligned}
\bar{V}(t_f) &= \phi(\bar{\mathbf{x}}(t_f), t_f) \\
\bar{V}_{\mathbf{x}}(t_f) &= \phi_{\mathbf{x}}(\bar{\mathbf{x}}(t_f), t_f) \\
\bar{V}_{\mathbf{xx}}(t_f) &= \phi_{\mathbf{xx}}(\bar{\mathbf{x}}(t_f), t_f)
\end{aligned}
\tag{64}
$$

## C   CONVERGENCE

We begin our convergence analysis for GTSONO with the definition of the global saddle point, which will prompt us to define the local saddle points.

**Definition C.1.** A point $(\mathbf{u}^\star, \mathbf{v}^\star) \in \mathbb{R}^{2n}$, is a global saddle point of a function $Q(\cdot, \cdot)$, if we have $f(\mathbf{u}^\star, \mathbf{v}) \leq f(\mathbf{u}^\star, \mathbf{v}^\star) \leq f(\mathbf{u}, \mathbf{v}^\star), \forall \mathbf{u} \in \mathbb{R}^n$, and $\mathbf{v} \in \mathbb{R}^n$.

For convenience on handling the variables we further define the vector $\mathbf{z} = [\mathbf{u}; \mathbf{v}] \in \mathbb{R}^{2n}$, along with the function $G: \mathbb{R}^{2n} \to \mathbb{R}^{2n}$, as $G(\mathbf{z}) = \begin{bmatrix}\nabla_{\mathbf{u}}Q(\mathbf{u}, \mathbf{v}) \\ -\nabla_{\mathbf{v}}Q(\mathbf{u}, \mathbf{v})\end{bmatrix}$. Accordingly the Jacobian of G is

given by: $JG(\mathbf{z}) = \begin{bmatrix} Q_{\mathbf{uu}} & Q_{\mathbf{uv}} \\ -Q_{\mathbf{vu}} & -Q_{\mathbf{vv}} \end{bmatrix} \in \mathbb{R}^{2n \times 2n}$. For the inverse of the Jacobian using the Schur complement Liu et al. (2021b), we obtain:

$$JG^{-1} = \begin{bmatrix} \tilde{Q}_{\mathbf{uu}}^{-1} & \tilde{Q}_{\mathbf{uu}}^{-1} Q_{\mathbf{uv}} Q_{\mathbf{vv}}^{-1} \\ -\tilde{Q}_{\mathbf{vv}}^{-1} Q_{\mathbf{vu}} Q_{\mathbf{uu}}^{-1} & -\tilde{Q}_{\mathbf{vv}}^{-1} \end{bmatrix} \tag{65}$$

where $\tilde{Q}_{\mathbf{uu}} = Q_{\mathbf{uu}} - Q_{\mathbf{uv}} Q_{\mathbf{vv}}^{-1} Q_{\mathbf{vu}}$, and similarly $\tilde{Q}_{\mathbf{vv}} = Q_{\mathbf{vv}} - Q_{\mathbf{vu}} Q_{\mathbf{uu}}^{-1} Q_{\mathbf{uv}}$. At this point we move to the update rule using the inverse of $JG$ as follows

$$\begin{bmatrix} \mathbf{u}_{k+1} \\ \mathbf{v}_{k+1} \end{bmatrix} = \begin{bmatrix} \mathbf{u}_k \\ \mathbf{v}_k \end{bmatrix} - \eta \begin{bmatrix} \tilde{Q}_{\mathbf{uu}}^{-1} & \tilde{Q}_{\mathbf{uu}}^{-1} Q_{\mathbf{uv}} Q_{\mathbf{vv}}^{-1} \\ -\tilde{Q}_{\mathbf{vv}}^{-1} Q_{\mathbf{vu}} Q_{\mathbf{uu}}^{-1} & -\tilde{Q}_{\mathbf{vv}}^{-1} \end{bmatrix} \begin{bmatrix} \nabla_{\mathbf{u}} Q(\mathbf{u}, \mathbf{v}) \\ -\nabla_{\mathbf{v}} Q(\mathbf{u}, \mathbf{v}) \end{bmatrix} \tag{66}$$

which yields the update rule for the open loop Min-Max DDP shown in section 3. Using the definition $\mathbf{z} = [\mathbf{u}; \mathbf{v}]$, we can write Eq. (66) more compactly as $\mathbf{z}_{k+1} \leftarrow \mathbf{z}_k - \eta JG^{-1}(\mathbf{z}_k)G(\mathbf{z}_k)$.

**Definition C.2.** Let $K_\gamma = \{\mathbf{z} \mid \|\mathbf{z} - \mathbf{z}^\star\| \leq \gamma\}$, be a neighborhood around fixed point $\mathbf{z}^\star = [\mathbf{u}^\star; \mathbf{v}^\star]$. Then, $\mathbf{z}^\star$ is a local saddle point, if $f(\mathbf{u}^\star, \mathbf{v}) \leq f(\mathbf{u}^\star, \mathbf{v}^\star) \leq f(\mathbf{u}, \mathbf{v}^\star), \forall \mathbf{u}, \mathbf{v} \in K_\gamma$.

Two fundamental assumptions regarding the Lipschitz continuity of $Q(\mathbf{u}, \mathbf{v})$ in region $K_\gamma$, along with the Lipschitz continuity of the Jacobian of $G(\mathbf{z})$, i.e., $JG(\mathbf{z})$, will enable us to derive the bounds for the iterates and function $Q$.

**Assumption C.3.** We assume that in this region, $Q$ is a $C^2$ function and that Q, and $JG$ are Lipschitz continous $\forall \mathbf{x}, \mathbf{y} \in K_\gamma$, i.e. we assume the following

$$\|Q(\mathbf{z}) - Q(\mathbf{z}^\star)\| \leq L' \|\mathbf{z} - \mathbf{z}^\star\| \tag{67}$$

$$\|JG(\mathbf{z}) - JG(\mathbf{z}^\star)\| \leq L \|\mathbf{z} - \mathbf{z}^\star\| \tag{68}$$

Additionally, consider constant $h$, for which it holds that: $\frac{1}{h} \geq \sigma_{\max}(JG^{-1})$, which is defined as the greatest singular value of inverse of the Jacobian of $G$. Now, using constants h and L, let us define the radius of region $K_\gamma$ as $\gamma < \frac{2h}{3L}$.

We proceeding by mentioning a Lemma which will be deemed helpful in the derivation of bounds for consecutive iterations.

**Lemma C.4** (Lemma 1 (Lesage-Landry et al. 2020)). *If $M \in \mathbb{R}^{n \times n}$ is invertible, then $\exists h > 0$, s.t.*

$$\|M^{-1}\| \leq \frac{1}{h} \iff \|M\mathbf{y}\| \geq h\|\mathbf{y}\|, \quad \forall \mathbf{y} \in \mathbb{R}^n. \tag{69}$$

In our analysis, we assumed a fixed point $\mathbf{z}^\star$, and local region $K_\gamma$ such that $\|\mathbf{z} - \mathbf{z}^\star\| \leq \gamma$, and followed the following steps:

1. Provide the sufficient conditions under which it is guaranteed that this fixed point is a local saddle point in region $K_\gamma$.

2. Show that our optimizer indeed converges to this fixed point, granted it is initialized within $K_\gamma$, which under the stipulation that the conditions in 4.2 are met has to be a local saddle point.

3. Show that our optimizer avoids unstable fixed points.

We start by stating the following Lemma.

**Lemma C.5** (Lemma 4, Adolphs et al. (2019)). *For a function Q, at least twice differentiable and not necessarily convex and not necessarily concave, a point $(\mathbf{u}^\star, \mathbf{v}^\star) \in K_\gamma$ is a local saddle point if the following conditions are met:* **Necessary condition:** $\nabla Q(\mathbf{u}^\star, \mathbf{v}^\star) = 0$, *and* **Sufficient condition:** $Q_{\mathbf{uu}} > 0$, *and* $Q_{\mathbf{vv}} < 0$

Intuitively, this Lemma expresses that $Q$ must be locally convex with respect to $\mathbf{u}$ and locally concave w.r.t. $\mathbf{v}$, at least in region $K_\gamma$ for $\mathbf{z}^\star$ to be a local saddle point. Then following Assumption 4.2, it holds that for appropriate selection of $R_{\mathbf{u}} > 0$ and $R_{\mathbf{v}} > 0$, we can ensure the positive definiteness of $Q_{\mathbf{uu}}$, and the negative definiteness of $Q_{\mathbf{vv}}$. It follows that if this condition is satisfied then it must also be true that $Q_{\mathbf{u}} = Q_{\mathbf{v}} = 0$, equivalently $G(\mathbf{z}^\star) = 0$. Now, we proceed with the following Lemma from which we will derive useful inequalities that will enable us to bound the iterates and the accumulative error of $Q$.

**Lemma C.6.** *Suppose the following conditions are met*

1. The initialization falls within a ball of radius $\gamma$ from $\mathbf{z}^\star$, namely $\mathbf{z}_0 \in K_\gamma$

2. Assumption 4.2 is satisfied, implying that $\mathbf{z}^\star$ is a local saddle, and thus $G(\mathbf{z}^\star) = 0$.

3. [Lemma 1; Lesage-Landry et al. (2020)]: For square and invertible matrix $JG$, $\exists h > 0$, such that $||JG^{-1}(\mathbf{z}^\star)|| \leq \frac{1}{h}$

4. The Jacobian of $G(\mathbf{z})$ is $L$-Lipschitz: $||JG(\mathbf{z}) - JG(\mathbf{z}^\star)|| \leq L||\mathbf{z} - \mathbf{z}^\star||$

*Then the following inequalities emerge between two consecutive iterations*

$$||\mathbf{z}_{k+1} - \mathbf{z}^\star|| < ||\mathbf{z}_k - \mathbf{z}^\star|| \qquad\qquad ||\mathbf{z}_{k+1} - \mathbf{z}^\star|| < \frac{3L}{2h}||\mathbf{z}_k - \mathbf{z}^\star||^2 \qquad (70)$$

*Proof.* We begin the proof with the update rule, and subtract the fixed point $\mathbf{z}^\star$ from both sides.

$$
\begin{aligned}
\mathbf{z}_{k+1} &= \mathbf{z}_k - \eta JG^{-1}(\mathbf{z}_k)G(\mathbf{z}_k) \implies \\
\mathbf{z}_{k+1} - \mathbf{z}^\star &= \mathbf{z}_k - \mathbf{z}^\star - \eta JG^{-1}(\mathbf{z}_k)G(\mathbf{z}_k) \\
&= \mathbf{z}_k - \mathbf{z}^\star - \eta JG^{-1}(\mathbf{z}_k)G(\mathbf{z}_k) + \eta JG^{-1}(\mathbf{z}_k)G(\mathbf{z}^\star) \\
&= \mathbf{z}_k - \mathbf{z}^\star + \eta JG^{-1}(\mathbf{z}_k)(G(\mathbf{z}^\star) - G(\mathbf{z}_k))
\end{aligned}
\qquad (71)
$$

where we used the fact that $G(\mathbf{z}^\star) = 0$. By the fundamental theorem of calculus,

$$
\begin{aligned}
\mathbf{z}_{k+1} - \mathbf{z}^\star &= \mathbf{z}_k - \mathbf{z}^\star + \eta JG^{-1}(\mathbf{z}_k)\int_0^1 JG(\mathbf{z}_k + \tau(\mathbf{z}^\star - \mathbf{z}_k))(\mathbf{z}^\star - \mathbf{z}_k)d\tau \\
&= JG^{-1}(\mathbf{z}_k)JG(\mathbf{z}_k)(\mathbf{z}_k - \mathbf{z}^\star) + JG^{-1}(\mathbf{z}_k)\int_0^1 \eta JG(\mathbf{z}_k + \tau(\mathbf{z}^\star - \mathbf{z}_k))(\mathbf{z}^\star - \mathbf{z}_k)d\tau \\
&= JG^{-1}(\mathbf{z}_k)\int_0^1 JG(\mathbf{z}_k)(\mathbf{z}_k - \mathbf{z}^\star)d\tau + JG^{-1}(\mathbf{z}_k)\int_0^1 \eta JG(\mathbf{z}_k + \tau(\mathbf{z}^\star - \mathbf{z}_k))(\mathbf{z}^\star - \mathbf{z}_k)d\tau \\
&= JG^{-1}(\mathbf{z}_k)\int_0^1 \left(\eta JG\big(\mathbf{z}_k + \tau(\mathbf{z}^\star - \mathbf{z}_k)\big) - JG(\mathbf{z}_k)\right)(\mathbf{z}^\star - \mathbf{z}_k)d\tau
\end{aligned}
\qquad (72)
$$

Consider $||\eta JG(\mathbf{z}_k + \tau(\mathbf{z}^\star - \mathbf{z}_k))|| \leq ||JG(\mathbf{z}_k + \tau(\mathbf{z}^\star - \mathbf{z}_k))||, \forall \eta \in (0, 1]$. Therefore, one obtains the following inequality

$$
\begin{aligned}
||\mathbf{z}_{k+1} - \mathbf{z}^\star|| &\leq ||JG^{-1}|| \quad ||\int_0^1 (JG(\mathbf{z}_k + \tau(\mathbf{z}_k - \mathbf{z}^\star)) - JG(\mathbf{z}_k))(\mathbf{z}^\star - \mathbf{z}_k)d\tau|| \\
&\leq ||JG^{-1}|| \quad ||\int_0^1 \tau L||\mathbf{z}_k - \mathbf{z}^\star||^2 d\tau|| \\
&\leq ||JG^{-1}||\frac{L}{2}||\mathbf{z}_k - \mathbf{z}^\star||^2
\end{aligned}
\qquad (73)
$$

At this point we use Lemma C.4 to derive an upper bound on the norm of the inverse of the Jacobian of the operator $G$. For arbitrary vector $\mathbf{y} \in \mathbb{R}^n$

$$
\begin{aligned}
||JG(\mathbf{z}_k)\mathbf{y}|| &= ||JG(\mathbf{z}_k)\mathbf{y} + JG(\mathbf{z}^\star)\mathbf{y} - JG(\mathbf{z}^\star)\mathbf{y}|| \\
&\geq ||JG(\mathbf{z}^\star)\mathbf{y}|| - ||JG(\mathbf{z}_k)\mathbf{y} - JG(\mathbf{z}^\star)\mathbf{y}|| \\
&\geq ||JG(\mathbf{z}^\star)\mathbf{y}|| - ||JG(\mathbf{z}_k) - JG(\mathbf{z}^\star)|| \, ||\mathbf{y}|| \\
&\geq h||\mathbf{y}|| - L||\mathbf{z}_k - \mathbf{z}^\star|| \, ||\mathbf{y}|| \\
&= (h - L||\mathbf{z}_k - \mathbf{z}^\star||)||\mathbf{y}||
\end{aligned}
\qquad (74)
$$

Thus since $\exists \mathbf{y} \in \mathbb{R}^n$, such that $||JG(\mathbf{z}_k)\mathbf{y}|| \geq (h - L||\mathbf{z}_k - \mathbf{z}^\star||)||\mathbf{y}||$, from Lemma C.4 we have that

$$||JG^{-1}(\mathbf{z}_k)|| \leq \frac{1}{h - L||\mathbf{z}_k - \mathbf{z}^\star||} \qquad (75)$$

Substituting Eq. (75) into (73), we obtain

$$||\mathbf{z}_{k+1} - \mathbf{z}^\star|| \leq \frac{L}{2(h - L||\mathbf{z}_k - \mathbf{z}^\star||)}||\mathbf{z}_k - \mathbf{z}^\star||^2 \tag{76}$$

By construction, we have that $||\mathbf{z}_k - \mathbf{z}^\star|| < \frac{2h}{3L}$, which enables us to obtain the following two identities.

$$\begin{aligned} ||\mathbf{z}_{k+1} - \mathbf{z}^\star|| &\leq \frac{L||\mathbf{z}_k - \mathbf{z}^\star||}{2(h - L||\mathbf{z}_k - \mathbf{z}^\star||)}||\mathbf{z}_k - \mathbf{z}^\star|| \\ &< \frac{L\frac{2h}{3L}}{2(h - L\frac{2h}{3L})}||\mathbf{z}_k - \mathbf{z}^\star|| = ||\mathbf{z}_k - \mathbf{z}^\star|| \end{aligned} \tag{77}$$

Trivially from Eq. (77), by bounding only the denominator, we obtain

$$\begin{aligned} ||\mathbf{z}_{k+1} - \mathbf{z}^\star|| &\leq \frac{L}{2(h - L||\mathbf{z}_k - \mathbf{z}^\star||)}||\mathbf{z}_k - \mathbf{z}^\star||^2 \\ &< \frac{L}{2(h - L\frac{2h}{3L})}||\mathbf{z}_k - \mathbf{z}^\star||^2 = \frac{3L}{2h}||\mathbf{z}_k - \mathbf{z}^\star||^2 \end{aligned} \tag{78}$$

yielding Eq. (70). □

Now, we are in position to prove convergence to fixed point $\mathbf{z}^\star$.

## C.1   PROOF OF THEOREM 4.3

For completeness we restate Theorem 4.3.

**Theorem C.7.** *Suppose $\mathbf{z}_0 \in K_\gamma$. Then for $L$-Lipschitz function $Q(\mathbf{z})$, with also $L'$-Lipschitz Jacobian of $G(\mathbf{z})$, the accumulative difference between iterates and fixed point $\mathbf{z}^\star$, as well as the total error from $Q(\mathbf{z}^\star)$ can be bounded by a constant*

$$\sum_{k=0}^{K} ||\mathbf{z}_k - \mathbf{z}^\star|| \leq \mathcal{O}(1) \qquad\qquad \sum_{k=0}^{k} |Q(\mathbf{z}_k) - Q(\mathbf{z}^\star)| \leq \mathcal{O}(1) \tag{79}$$

*Proof.* Let us first show that $\sum_{k=1}^{K} ||\mathbf{z}_t - \mathbf{z}^\star||$ is bounded. By using the second identity obtained by the previous lemma we obtain

$$\sum_{k=1}^{K} ||\mathbf{z}_k - \mathbf{z}^\star|| < \sum_{k=1}^{K} ||\mathbf{z}_{k-1} - \mathbf{z}^\star||^2 \frac{3L}{2h} \tag{80}$$

We want to make the RHS of the inequality to have the same index as the LHS

$$\sum_{k=1}^{K} ||\mathbf{z}_{k-1} - \mathbf{z}^\star||^2 \frac{3L}{2h} = \sum_{k=1}^{K} ||\mathbf{z}_k - \mathbf{z}^\star||^2 \frac{3L}{2h} + \frac{3L}{2h}(e_0 - e_K) \tag{81}$$

where $e_0 = ||\mathbf{z}_0 - \mathbf{z}^\star||^2$, and $e_K = ||\mathbf{z}_K - \mathbf{z}^\star||^2$ are the individual error term at the initialization phase and the $K^{\text{th}}$ iteration respectively. From this definition, it follows that $e_0, e_K > 0$, hence $\sum_{k=1}^{K} ||\mathbf{z}_k - \mathbf{z}^\star||^2 \frac{3L}{2h} + \frac{3L}{2h}(e_0 - e_K) \leq \sum_{k=1}^{K} ||\mathbf{z}_k - \mathbf{z}^\star||^2 \frac{3L}{2h} + \frac{3L}{2h}e_0$. Moreover, we observe that in the RHS we have $||\mathbf{z}_k - \mathbf{z}^\star||^2$, using the assumption that we fall into the region $K_\gamma$, where $||\mathbf{z} - \mathbf{z}^\star|| \leq \gamma$ yields

$$\sum_{k=1}^{K} ||\mathbf{z}_k - \mathbf{z}^\star|| \leq \sum_{k=1}^{K} ||\mathbf{z}_k - \mathbf{z}^\star|| \frac{3L\gamma}{2h} + \frac{3L}{2h}e_0 \tag{82}$$

Therefore, we have

$$\sum_{k=1}^{K} ||\mathbf{z}_k - \mathbf{z}^\star|| < \sum_{k=1}^{K} ||\mathbf{z}_k - \mathbf{z}^\star|| \frac{3L\gamma}{2h} + \frac{3L}{2h} e_0$$

$$\sum_{k=1}^{K} (1 - \frac{3L\gamma}{2h}) ||\mathbf{z}_k - \mathbf{z}^\star|| < \frac{3L}{2h} e_0 \tag{83}$$

$$\sum_{k=1}^{K} ||\mathbf{z}_k - \mathbf{z}^\star|| < \frac{\delta}{(1 - \frac{3L\gamma}{2h})}$$

where $\delta = \frac{3L}{2h} e_0$. In the last inequality we can divide with $1 - \frac{3L\gamma}{2h}$, since from our assumption $\gamma < \frac{2h}{3L}$. Therefore, since $\lim_{K\to\infty} \sum_{k=0}^{K} ||\mathbf{z}_k - \mathbf{z}^\star|| < \frac{\delta}{(1 - \frac{3L\gamma}{2h})}$ is bounded by a constant and $||\mathbf{z}_k - \mathbf{z}^\star|| < ||\mathbf{z}_{k-1} - \mathbf{z}^\star||$, it shows that $||\mathbf{z}_k - \mathbf{z}^\star|| \to 0$, for $k \to \infty$.
Now, recall that the function $Q(\mathbf{z})$ is also $L'$-Lipschitz at point $\mathbf{z}^\star$, thus

$$\sum_{k=1}^{K} |Q(\mathbf{z}_k) - Q(\mathbf{z}^\star)| \le \sum_{k=1}^{K} L' ||\mathbf{z}_k - \mathbf{z}^\star|| \tag{84}$$

Hence, readily by combining the last two inequalities we obtain

$$\sum_{k=1}^{K} |Q(\mathbf{z}_k) - Q(\mathbf{z}^\star)| < L' \frac{\delta}{(1 - \frac{3L\gamma}{2h})} = O(1) \tag{85}$$

$\square$

This implies that the iterates converge and the total error goes to 0, as the number of iterations increases to infinity. We proceed to show that if our algorithm converges to a strictly stable, implying local convergence (Wang et al. (2019)). Recall that for fixed point $z^\star = (\mathbf{u}^\star, \mathbf{v}^\star)$, for which it holds $\nabla f(z^\star) = 0$ if the Jacobian of the update rule satisfies $\rho(J) \le 1$, then this point is assymptotically locally stable [Proposition 1.4 (Daskalakis & Panageas (2018))]. The Jacobian of our update rule is given by

$$J = I - \eta([JG]^{-1}[JG] + \nabla(JG^{-1})F) \tag{86}$$

## C.2 PROOF OF PROPOSITION 4.5

We restate the Proposition for completeness.

**Proposition C.8.** *Considering linearized ODE dynamics, we can deduce that for the derivatives $Q_{\mathbf{uu}}$ and $Q_{\mathbf{vv}}$, with respect to $\mathbf{z} = [\mathbf{u}, \mathbf{v}]$: $\frac{\partial}{\partial \mathbf{z}} Q_{\mathbf{uu}} = \frac{\partial}{\partial \mathbf{z}} Q_{\mathbf{vv}} = 0$. This implies that $\nabla(JG^{-1}) = 0$.*

*Proof.* We begin with considering the linear approximation of non-linear dynamics along the nominal trajectory, an approximation upon which the entire DDP scheme is based Pan & Theodorou (2014). This enables us to set second order terms with respect to the dynamics equal to zero, namely $F_{\mathbf{uu}} = F_{\mathbf{vv}} = F_{\mathbf{uv}} = F_{\mathbf{xu}} = F_{\mathbf{xv}} = F_{\mathbf{xx}} = 0$. Note that this approximation was also held to derive the ODEs in Eq. (23).

**Proposition C.9.** *Recall the ODE we derived for the decomposing $Q_{\mathbf{xx}}$, $\mathbf{q}_i$ for $i = \{1, \ldots, R \le m\}$ $-\frac{d\mathbf{q}_i(t)}{dt} = F_{\mathbf{x}} \mathbf{q}_i(t)$ with terminal condition $\mathbf{q}_i(t_f) = \mathbf{y}_i$. We can express $\mathbf{q}_i$ as*

$$\mathbf{q}_i(t) = \mathbf{y}_i \exp\left(\int_t^{t_f} -F_{\mathbf{x}} d\tau\right) \propto \exp\left(\int_t^{t_f} -F_{\mathbf{x}} d\tau\right) \tag{87}$$

Using this expression for $\mathbf{q}_i(t)$, we can calculate its partial derivatives with respect to $\mathbf{x}, \mathbf{u}$, and $\mathbf{v}$.

$$\frac{\partial \mathbf{q}_i}{\partial \mathbf{x}} = \left(\int_t^{t_f} -F_{\mathbf{xx}} d\tau\right) \exp\left(\int_t^{t_f} -F_{\mathbf{x}} d\tau\right) = (C_1 - C_1) \mathbf{q_i}(t) = 0 \tag{88}$$

Similarly for the other derivatives,

$$
\begin{aligned}
\frac{\partial \mathbf{q}_i}{\partial \mathbf{u}} &= \int_t^{t_f} -F_{\mathbf{xu}} d\tau \exp\left(\int_t^{t_f} -F_{\mathbf{x}} d\tau\right) = 0 \\
\frac{\partial \mathbf{q}_i}{\partial \mathbf{v}} &= \int_t^{t_f} -F_{\mathbf{xv}} d\tau \exp\left(\int_t^{t_f} -F_{\mathbf{x}} d\tau\right) = 0
\end{aligned}
\tag{89}
$$

At this point, we can easily move to the expession for the Hessians in Eq. 35, and see that differentation of these expressions with respect to any set of weights results in 0.

$$
\begin{aligned}
\frac{\partial Q_{\mathbf{uu}}}{\partial \mathbf{u}} &= \frac{\partial}{\partial \mathbf{u}}\left[\left(\int F_{\mathbf{u}} \mathbf{q}_i\right)\left(\int F_{\mathbf{u}} \mathbf{q}_i\right)^{\mathsf{T}}\right] \\
&= \left(\int F_{\mathbf{uu}} \mathbf{q}_i + F_{\mathbf{u}} \mathbf{q}_{i,\mathbf{u}}\right)\left(\int F_{\mathbf{u}} \mathbf{q}_i\right)^{\mathsf{T}} + \left(\int F_{\mathbf{u}} \mathbf{q}_i\right)\left(\int F_{\mathbf{uu}} \mathbf{q}_i + F_{\mathbf{u}} \mathbf{q}_{i,\mathbf{u}}\right)^{\mathsf{T}} \\
&= 0
\end{aligned}
\tag{90}
$$

Similarly, it follows for the derivative of every second order matrix with respect to $\mathbf{u}$, or $\mathbf{v}$ resulting in $JG^{-1}$. $\qquad\square$

From this proposition, we can easily infer from Eq. (86) that $||J|| < 1, \forall \eta \in (0,1)$. Therefore, it holds that the spectral norm of the Jacobian to be less than 1 and ensuring the stability, showing that our optimizer convergenges to stable saddle points.

## D  EXPERIMENT DETAILS

**Networks and ODE Solver**  The ODE solver we used for every experiment is the standard Runge-Kutta 4(5) adaptive solver (dopri5; Dormand & Prince (1980)) implemented by the torchdiffeq package, with the numerical tolerance set to 1e-3, and fixed integration time $[0, 1]$ for all experiments. All experiments are conducted on the same GPU machine (TITAN RTX) and implemented with pytorch

**Model configuration**  Here, we specify the model We will adopt the following syntax to describe the layer configuration.

- Conv(input, output, kernel, stride)
- FC(input, output)

Using this notation, Table 6 presents the architecture of every model.

Table 6: Network Structure

| Network |
| --- |
| **Discrete Convolution Layers**: ReLU(Conv(64, 64, 3, 1)) $\rightarrow$ ReLU(Conv(64, 64, 3, 1)) $\rightarrow$ Conv(64, 64, 3, 1) |
| **Continuous Convolution Layers**: ReLU(Conv(64, 65, 3, 1)) |
| **Feed Forward Layers**: ReLU(FC(2304, 500)) $\rightarrow$ FC(500, 10) |

Note that for our optimizers, a game theoretic layer would be implemented as ReLU(Layer$_{\mathbf{u}}$ + Layer$_{\mathbf{v}}$). Recall that in GTSONO every layer has a game theoretic counterpart and in c-GTSONO we have only convolution game theoretic layers.

**Dataset**  Recall that the datasets examined for our experiments are the CIFAR10, and SVHN. Both image datasets are preprocessed with standardization. CIFAR10 contains 50000 training images and 10000 test images, whereas SVHN contains 73257 digits for training and 26032 for testing. PGD and FGSM require computation of the loss gradient. For this reason, to facilitate and accelerate testing against adversarial attacks, we utilized 40% of the samples in the test set of CIFAR10, and 20% of the test set on SVHN.

**Tuning process**   For the comparison among the optimizers, the tuning varied for each optimizer. More specifically, for Adam and SGD only the learning rate was evaluated. For SNOpt, we tuned its learning rate and its regularization term, while the 2 variations of GTSONO were tuned against its learning rate, the regularization constant for the two sets of weights. We perform a grid search for the tuning of the hyper-parameters for every optimizer. The tuning process took place by optimizing the performance of each optimizer in the clean test set and evaluating its robustness when convergence was noticed. Robust overfitting was observed when evaluating the robustness of the models many iterations after convergence, regardless of the optimizer. Table 7 summarizes the hyperparameters evaluated for each optimizer, along with the examined values.

Table 7: Hyperparameters for each optimizer and their examined values

| Optimizer | Hyperparameter | Value |
|---|---|---|
| Adam, SGD | Learning rate | $\{1e^{-4}, 2.5e^{-4}, 5e^{-4}, 7.5e^{-4}, 1e^{-3}, 2.5e^{-3}, 5e^{-3}, 7.5e^{-3}, 1e^{-2}\}$ |
| SNOpt | Learning rate, Regularization constant | $\{1e^{-4}, 2.5e^{-4}, 5e^{-4}, 7.5e^{-4}, 1e^{-3}, 2.5e^{-3}, 5e^{-3}, 7.5e^{-3}, 1e^{-2}\}$ $\{1e^{-2}, 2.5e^{-2}, 5e^{-2}\}$ |
| *GTSONO* | Learning rate, Regularization constants $R_{\mathbf{u}}$, $R_{\mathbf{v}}$ | $\{1e^{-4}, 2.5e^{-4}, 5e^{-4}, 7.5e^{-4}, 1e^{-3}, 2.5e^{-3}, 5e^{-3}, 7.5e^{-3}, 1e^{-2}\}$ $\{1e^{-4}, 2.5e^{-4}, 5e^{-4}, 1e^{-3}, 2.5e^{-3}, 5e^{-3}, 1e^{-2}, 2.5e^{-2}, 5e^{-2}\}$ |

**Attacks**   Recall that in this study, we focused on the white box $\ell_\infty$-norm Projected Gradient Descent, Fast Gradient Sign Method attacks and gray/ black-box attack CW attack to assess the robustness of our models. For $\ell_\infty$ PGD attack, the update rule to generate adversarially attacked examples is given as follows

$$\mathbf{x}' \leftarrow \Pi_{\mathbb{B}_\infty}(\mathbf{x} + \eta_1 \, \text{sign}(\nabla_{\mathbf{x}'}\mathcal{L}(F(\mathbf{x}'), y)))$$

, where $\Pi_{\mathbb{B}_\infty}$ is the projection operator on the $\ell_\infty$ norm.

For the FGSM attack, the update rule to generate adversarially attacked examples is given as follows

$$\mathbf{x}' \leftarrow \mathbf{x} \, + \, \eta_2 \, \text{sign}(\nabla_{\mathbf{x}}\mathcal{L}(F(\mathbf{x}), y)))$$

For the update rule to generate adversarially attacked examples through CW, we refer the interested reader to Carlini & Wagner (2017).

**Adversarial Training Methods**   For the second round of our experiments we explore the versatility of GTSONO in its ability to be efficiently adapted to other adversarial training methods such as the Free Adversarial Training(FreeAT) scheme (Shafahi et al. (2019)) and the TRADES objective function Zhang et al. (2019b). In both of these frameworks, SGD optimizer was employed as the standard optimizer.

More explicitly, FreeAT proposed the modification of training each minibatch *m* times providing strong adversarial examples, while also using the perturbation from the previous minibatch to warmstart the perturbation for the new minibatch. This is the hyperparameter we performed our ablation analysis, by setting *m* equal to 4 and 8. Algorithm 2 summarizes the FreeAT method.

From our results, it was shown that GTSONO was able to outperform the benchmark optimizer in this adversarial training method, and in less epochs resulting in an overall faster training time, leveraging the faster convergence of our second order method, thus rendering it a more efficient and effective method for this adversarial training scheme.

Furthermore, TRADES proposed a new objective function which computes the KL divergence, between natural and adversarial images, multiplied with a constant $1/\lambda$. This is the hyperparameter against which the ablated analysis. Algorithm 3 summarizes the TRADES formulation. It was again demonstrated that GTSONO was able not only to outperform the benchmark optimizer in terms of raw performance but it was again shown that it is the more efficient option for this adversarial training method, as it provided superior performance in less epochs, reducing significant the overall training time.

| **Algorithm 2** Free Adv. Training Algorithm | **Algorithm 3** TRADES Algorithm |
|---|---|
| 1: **Input**: Samples $X \sim D$, perturbation $\epsilon$, training steps $m$, learning rate $\eta$ 
 2: Initialize $\theta, \delta = 0$ 
 3:  **for** epoch = 1 .. N 
 4:    **for minibatch** $B \subset X$ 
 5:      **for** i = 1 .. m 
 6:        $g_\theta \leftarrow \mathbb{E}_{(\mathbf{x},\mathbf{y}) \in B}[\nabla_\theta \ell(\mathbf{x} + \delta, \mathbf{y}; \theta)]$ 
 7:        $g_{\text{adv}} \leftarrow \nabla_{\mathbf{x}} \ell(\mathbf{x} + \delta, \mathbf{y}; \theta)$ 
 8:        Update $\theta, \theta \leftarrow \theta - \eta g_\theta$ 
 9:        Calculate $\delta, \delta \leftarrow \delta + \epsilon\,\text{sign}(g_{\text{adv}})$ 
 10:        $\delta \leftarrow \text{clip}(\delta, -\epsilon, \epsilon)$ 
 11:      **end for** 
 12:    **end for** 
 13: **end for** | 1: **Input**: Step sizes $\eta_1, \eta_2$, batch size $m$, number of iterations $m$ in inner optimization, perturbation $\epsilon$, network with parameters $\theta$ 
 2: **repeat** 
 3:    Read from minibatch $B = \{\mathbf{x}_1, \ldots, \mathbf{x}_b\}$ 
 4:    **for** i = 1 .. b 
 5:      $\mathbf{x}_i \leftarrow \mathbf{x}_i + 0.001N(0, I)$, where $N(O, I)$ is the Gaussian distribution with 0 mean and identity variance 
 6:      **for** k = 1 .. m 
 7:        $\mathbf{x}_i' \leftarrow \Pi_{\mathbf{x}_i, \epsilon}(\eta_1\text{sign}(\nabla_{\mathbf{x}_i'} L(f_\theta(\mathbf{x}_i'), \mathbf{x}_i))) + \mathbf{x}_i$ 
 8:      **end for** 
 9:    **end for** 
 10:    $\theta \leftarrow \eta_2\nabla_\theta[L(f_\theta(\mathbf{x}_i), \mathbf{y}_i) + L(f_\theta(\mathbf{x}_i), f_\theta(\mathbf{x}_i'))/\lambda]\frac{1}{b}$ 
 11: **until** converges |

For our experiments, we selected the perturbation distance $\epsilon$ to be equal to 0.03 in both ablation studies. The number of internal iteration in FreeAT is the hyperparameter we performed our ablation analysis, by setting $m$ equal to 4 and 8. In our adaptation of GTSONO in TRADES, we set the iternal iterations to be equal to 5, and we performed ablation study with respect to constant $\lambda$.

**Limitations** As shown in the Tables above, the achieved robustness of our optimizer comes at the price of a slower per iteration training time and higher memory consumption. This is attributed to the decomposition and calculation of the three matrices involved in the open loop min-max DDP update $Q_{\mathbf{uu}}, Q_{\mathbf{uv}}, Q_{\mathbf{vv}}$. Although, memory consumption indicated in Tables 8-12 suggest that GTSONO is certainly affordable for current state-of-the-art GPUs, its requirements for memory usage and its greater training-per-iteration times motivate us to experiment with recasting the proposed optimizer through distributed optimization distributed DDP schemes.

Additionally, another shortcoming noticed not only in our method by in general in every neural ODE model was the robust overfitting. More specifically, after convergence was observed, even for stagnant natural accuracy, we observed that their robust accuracy decreased as the number of epochs increased. This naturally creates future lines of work of experimenting with more network structures and types, as well as regularization techniques to attempt to reduce this robust overfitting.

**Training Time and Memory Consumption** At this point we compare the resources required by each optimizer.

1. **Optimizer Comparison:** For this round of experiments, we compared optimizers in the task of image classification, both in the natural test set and recorded their natural accuracy but also their robust accuracy under various attacks, such as FGSM and PGD for various degrees of perturbation. Every optimizer was trained with a batch size of 500, and for 15 epochs. Tables 8 and 9 present the time and consummed memory by each optimizer on CIFAR10 and SVHN respectively.

Table 8: Training time and computational resources for Optimizers on CIFAR10

| Optimizer | Parameters ($10^6$) | Total training time (min : sec) | Memory Consumption (GB) |
|---|---|---|---|
| Adam | 1.25 | 2:53 | **2.13** |
| SGD | 1.25 | **2:37** | **2.13** |
| SNOpt | 1.25 | 3:01 | 4.58 |
| *GTSONO* | 2.51 | 4:01 | 6.71 |
| *C-GTSONO* | 1.35 | 3:53 | 6.71 |

Table 9: Training time and computational resources for Optimizers on SVHN

| Optimizer | Parameters $(10^6)$ | Total training time (min : sec) | Memory Consumption (GB) |
|---|---|---|---|
| Adam | 1.25 | 4:13 | **2.13** |
| SGD | 1.25 | **4:09** | **2.13** |
| SNOpt | 1.25 | 4:23 | 5.50 |
| *GTSONO* | 2.51 | 6:05 | 6.71 |
| *C-GTSONO* | 1.35 | 5:22 | 6.71 |

2. **Adaptation to adversarial training methods:** As mentioned previously, we evaluate the robustness of each optimizer in each measurement after convergence has been observed. Figure 3 illustrates the faster convergence of GTSONO compared to SGD in both defense schemes. More specifically, taking into consideration the graphs in Figure 3, we list the epochs for which each optimizer was trained.

   - FreeAT $m = 4$: SGD 15 epochs GTSONO 10 epochs
   - FreeAT $m = 8$: SGD 40 epochs GTSONO 5 epochs
   - TRADES $\lambda = 1/6$: SGD 20 epochs GTSONO 10 epochs
   - TRADES $\lambda = 1/10$: SGD 20 epochs GTSONO 10 epochs

Table 10: Training time and computational resources for FreeAT, for $m = 4$

| Optimizer | Parameters $(10^6)$ | Training time per iteration (sec) | Total training time (min:sec) | Memory Consumption (GB) |
|---|---|---|---|---|
| SGD | 1.25 | **0.53** | 13:15 | **2.13** |
| *C-GTSONO* | 1.35 | 0.70 | **11:39** | 6.09 |

Table 11: Training time and computational resources for FreeAT, for $m = 8$

| Optimizer | Parameters $(10^6)$ | Training time per iteration (sec) | Total training time (min:sec) | Memory Consumption (GB) |
|---|---|---|---|---|
| SGD | 1.25 | **0.63** | 64:45 | **2.13** |
| *C-GTSONO* | 1.35 | 1.1 | **8:21** | 6.09 |

Table 12: Training time and memory resources for TRADES

| Optimizer | Parameters $(10^6)$ | Training time per iteration (sec) | Total training time (min:sec) | Memory Consumption (GB) |
|---|---|---|---|---|
| SGD | 1.25 | **0.92** | 30:30 | **3.48** |
| *C-GTSONO* | 1.35 | 1.39 | **22:18** | 7.61 |

