# OpenReview forum: "A ROBUST DIFFERENTIAL NEURAL ODE OPTIMIZER"
_ICLR.cc/2024/Conference — ICLR 2024 poster_

### Official Review · Reviewer_gukw · 2023-10-30

**Soundness:** 3 good
**Presentation:** 2 fair
**Contribution:** 3 good
**Rating:** 6
**Confidence:** 3

**Summary:**

The paper introduces a robust optimization algorithm called GTSONO, designed to train Neural Ordinary Differential Equations (Neural ODEs) that are more resilient to adversarial attacks. Based on min-max Differential Dynamic Programming, the algorithm is not only computationally efficient but also guarantees convergence to local saddle points. Experimental results demonstrate its significant advantage in improving model robustness compared to benchmark optimizers. Overall, the paper offers a new and effective tool for enhancing the robustness of deep learning models.

**Strengths:**

Originality:
The paper introduces a novel perspective by interpreting Neural ODE optimization as a min-max optimal control problem. GTSONO's approach, rooted in min-max Differential Dynamic Programming, showcases a creative amalgamation of existing concepts.
Quality:
Offering convergence guarantees to local saddle points signifies the robustness of GTSONO. Efficient matrix decompositions and strong empirical results further attest to the research's high quality.
Clarity:
The document articulately bridges intricate theoretical concepts with empirical findings. Despite some formatting issues, the presentation remains clear and coherent.
Significance:
This work addresses a pivotal challenge in neural ODEs, enhancing their robustness against adversarial attacks. The exploration of optimal control paradigms in adversarial training methods underscores its contributions in the domain.

**Weaknesses:**

1. Although the GTSONO optimizer proposed in this paper can improve the robustness of the model to some extent, it adds too much extra computational overhead, which is unacceptable in the training of larger models.
2. The paper's limited comparison with just one other optimizer that improves robustness diminishes the persuasiveness of the results, as a more comprehensive comparison would have strengthened the findings.
3. One limitation of this paper is that the experiments are confined to the CIFAR10 and SVHN datasets, with no validation on more extensive datasets such as CIFAR100 and ImageNet. This restricts the applicability of the research to a certain extent.

**Questions:**

1. Could the authors provide insights into the feasibility of applying GTSONO to larger and more complex datasets such as CIFAR100 and ImageNet?
2. Given the focus on robustness improvement, could the authors consider including a more extensive comparison with various state-of-the-art optimizers that enhance robustness?
3. Are there any further insights into the stability and convergence properties of GTSONO, especially under varying hyperparameters and different neural network architectures?
4. In general, deep learning optimizers have used mini-batch size to obtain a partial batch of datasets, which means that random noise will be introduced, so the optimizer's parameter update rule should correspond to a stochastic differential equation instead of an ordinary differential equation. Therefore, why do the authors use an ODE rather than an SDE for their theoretical analysis?

---

> ### Author Response · Authors · 2023-11-18
> **Author Response to Reviewer gukw**
>
> We would like to thank the reviewer for their positive remarks, as well as their useful comments and further suggestions for improving our work. In the following we address each of the concerns and suggestions of the reviewer.
>
> ### **1.  Computational overhead**
>
> - While we admit that our method introduces a second set of weights, we highlight that most adversarial training methods typically include nested optimization schemes that are computationally demanding. The main advantage of our methodology is that the antagonizing weights  are updated through a *single* backpropagation, which reduces computational complexity while also achieving robustness even with natural training bypassing the computational burden of perturbing the input.  In addition, note that we also enhance computational efficiency by exploiting efficient matrix decompositions.
>
> ---
>
> ### **2.Experiments on CIFAR-100 and Tiny ImageNet and a more extensive comparison with various state-of-the-art optimizers that enhance robustness**
> - Following the suggestions of the reviewer, we present further comparisons against the state-of-the-art optimizers that enhance robustness such as Lyanet[1], and FI-ODE[2]. Additionally, we also evaluate the all optimizers in additional datasets: CIFAR100 and TinyImageNet. These results in the Table below further indicate the robustness of our optimizer as it outperforms state-of-the-art robust optimizers in multiple datasets.  We also highlight the significantly shorter training time required by GTSONO in contrast to Lyanet and FI-ODE.
>
>     | Optimizer | CIFAR100-PGD_20^0.03 | Tiny-ImageNet PGD_20^0.03 | Training TinyImageNet (min:sec) | # of Parameters |
>     | --- | --- | --- | --- | --- |
>     | Adam | $10.7\pm1.1$ | $1.9\pm0.1$ | 5:37 | 1.35 M |
>     | SNOpt | $19.7\pm0.6$ | $3.6\pm0.9$ | 5:41 | 1.35 M |
>     | SGD | $9.6\pm0.5$ | $0.8\pm0.2$ | 5:20 | 1.35 M |
>     | LyaNet^* | $20.1\pm0.5$ | $5.4\pm0.4$ | 313:21 | 19.6 M |
>     | FI-ODE^* | $15.4\pm0.4  $ | $4.0\pm0.2$ | 607:56 | 2.4 M |
>     | c-GTSONO | $23.5\pm0.1$ | $9.1\pm0.8$ | 6:35 | 1.47M |
>
>     | CIFAR10-Optimizer | PGD_20^0.03 | PGD_20^0.05 | Time (min:sec) | # of Parameters |
>     | --- | --- | --- | --- | --- |
>     | LyaNet^* | $46.1\pm0.4$ | $32.7\pm0.6$ | 32:44 | 19.6 M |
>     | FI-ODE^* | $48.5\pm0.3$ | $33.4\pm0.6$ | 143:17 | 2.4 M |
>     | c-GTSONO | $50.6 ± 0.3$| $35.0 \pm 0.2$ | 3:53 | 1.35M |
>
> ---
>
> ### **3. Insights into the stability and convergence properties of GTSONO, especially under varying hyperparameters and different neural network architectures?**
>
> - The convergence guarantees are agnostic of the architecture and hyperparameter configuration. The only necessary assumption for the convergence guarantees to hold should be that the function of our accumulated loss $Q$ is locally convex-concave - which can be imposed through appropriate selection of the regularization parameters $R_u, R_v$
>
> ---
>
> ### **4. In general, deep learning optimizers have used mini-batch size to obtain a partial batch of datasets, which means that random noise will be introduced, so the optimizer's parameter update rule should correspond to a stochastic differential equation instead of an ordinary differential equation. Therefore, why do the authors use an ODE rather than an SDE for their theoretical analysis?**
>
> - We would like to kindly emphasize that the stochasticity comes from the input, however this does not affect the dynamics which remain deterministic. Therefore, the appropriate representation is through an ODE instead of an SDE. This allows us to employ deterministic optimal control which naturally leads to a deterministic update law.

---

> ### Author Response · Authors · 2023-11-21
> **Follow up to Reviewer gukw**
>
> As we are approaching the end of author-reviewer discussion, we would appreciate the reviewer to clarify if our responses has sufficiently addressed the raised concerns, and, if so, we kindly appreciate the reviewer to re-consider their rating.

---

### Official Review · Reviewer_L7bT · 2023-11-01

**Soundness:** 3 good
**Presentation:** 2 fair
**Contribution:** 3 good
**Rating:** 6
**Confidence:** 3

**Summary:**

This paper interprets neural ODE optimization as a min-max optimal control problem, and proposed a second order optimizer. The proposed optimizer is computationally feasible by matrix decomposition. Empirically, the authors compare with other optimizers (Adam, SGD and another not min-max second order optimizer), and show it has improved adversarial robustness.

**Strengths:**

This paper proposes an interesting perspective of robust neural ODE optimization via min-max optimal control.

The authors make the proposed second order optimizer to be computational feasible: instead of back propagate coupled matrices, one can back propane a set of vectors.

The authors provide convergence guarantee of the proposed optimizer.

Experiments show improved adversarial robustness comparing with other non-robust neural ODE optimizers.

**Weaknesses:**

1. Although I think it is interesting to formulate robust neural ODE optimization as min-max OC, I feel there is gap why the formulation in (3) can be beneficial to the robustness problem in (1): in (3), the adversary is on neural network weights, and in (1), the adversary is on the inputs to the neural network.

2. Despite the effort of reducing computational cost, the proposed method is still very expensive. It would be good to include a complexity comparison between other neural ODE optimizers: first-order adjoint and SNOpt.

3. It is known that neural ODE tends to suffer from gradient obfuscation issue when being evaluated for empirical adversarial robustness. It would be beneficial to have some adaptive attacks or non-gradient based attacks to make sure the improved robustness is valid. For instance, the attacks used in [1]. Since the optimizer on its own has lower robustness accuracy than adversarial training methods, it is crucial to have solid experiments to show its benefits when combining with other robust training techniques. In general, I like the min-max OC perspective, but it may still lack evidence for its usefulness. Maybe the authors could also consider evaluating against adversary on neural network weights, which I think is more close to the formulation.

[1] Kang, Qiyu, et al. "Stable neural ode with lyapunov-stable equilibrium points for defending against adversarial attacks." Advances in Neural Information Processing Systems 34 (2021): 14925-14937.

**Questions:**

1. My main question is as in weakness 1: the proposed optimizer seems to be beneficial to attacks on neural network weights rather than on the inputs. I hope the authors can clarify why they choose to demonstrate the effectiveness of their optimizer on input-robustness, and will the method be useful for attacks on system weights?

2. In the experiments, it seems that only having adversarial control on convolution layers is much better than having them on all of the layers. From the theory parts it is not clear why this is the case. The authors should provide more analysis on this.

3. When combing with adversarial training (table 4), why CW accuracy drops? This may indicate some gradient obfuscation issue, it will be good to include some stronger attacks in the evaluation (like AutoAttack, Square) as suggested in weakness 3.

---

> ### Author Response · Authors · 2023-11-18
> **Author Response to Reviewer L7bT**
>
> We truly appreciate the encouraging comments of the reviewer, as well as their useful remarks.  In the following we address the points raised by the reviewer.
>
> ### **1. Gap why the optimization of min-max optimal control can be beneficial to the robustness problem in the adversarial training problem**
>
> - In our approach, we introduce the antagonizing set of controls to represent the disturbances in the system, inspired by game theoretic optimal control. This key representation enables us to utilize computationally efficient and scalable trajectory optimizers such as min-max Differential Dynamic Programming (DDP). In the realm of Optimal Control (OC), it has been demonstrated that game-theoretic formulation enables the model handling uncertainties and external disturbances [1].
> - The key advantage of incorporating as adversarial control inputs lies in the fact that instead of typically obtaining nested optimization schemes that are computationally demanding, our algorithm updates the antagonizing weights with a *single* backward pass which reduces computational complexity while also achieving robustness even with natural training bypassing the need of perturbed input. **
>
> ---
>
> ### **2. Complexity comparison between GTSONO and other Neural ODE optimizers: first-order adjoint and SNOpt.**
>
> - We updated the paper, in order to present better the complexity comparison between GTSONO and other Neural ODE optimizers, adding a pertinent table in the main part of the paper. For an extensive comparison with regards to the computational requirement of the optimizers, we refer the reviewer to Tables 8-12 in Appendix D, which show training time and memory consumption.
>
> ---
>
>
> ### **3. Clarification on the selection to demonstrate the effectiveness of our optimizer on input-robustness**
>
> - While we mostly refer the reviewer in weakness 1, regarding our motivation to recast the optimization problem with input disturbance to a game theoretic formulation. We chose to demonstrate the effectiveness on input-robustness, given that this is the most widely used form of attack. We also appreciate the suggestions of the reviewer to include studies on systems weights which we intend to explore in our future work.
>
> ---
>
> ### **4. In the experiments, it seems that only having adversarial control on convolution layers is much better than having them on all of the layers. From the theory parts it is not clear why this is the case. The authors should provide more analysis on this.**
>
> - We provide some intuition for this phenomenon. To quantify the similarity between the feature vector of the models from attacked and clean images, we calculate the cosine similarity. We find that the model with adversary weights in the convolution layers evaluated with attacked images is significantly closer to the feature vector from the clean images (cosine_similarity=0.92), compared to the feature vector from attacked images in the model without adversary control in the convolution layers (cosine_similarity=0.21). Therefore, feeding a robust feature vector to the classifier is similar to feeding the feature vector from the clean images, which eliminates the need to include antagonizing weights in the last linear layers, as it is well known that they drop the natural accuracy of model. Therefore, since the disturbance is in the input, this implies that it suffices to add adversary control in the convolution layers in order to capture robust features for robustifying the model. We intend to to investigate more thoroughly the underlying theory in future work.
>
> ---
>
> ### **5. When combing with adversarial training (table 4), why CW accuracy drops? This may indicate some gradient obfuscation issue, it will be good to include some stronger attacks in the evaluation (like AutoAttack, Square) as suggested in weakness 3.**
>
> - Based on the reviewer’s recommendations we have also added AutoAttack and Square attacks to assess our optimizer.    In the tables below, we compare the robustness of GTSONO against the baseline classifiers on CIFAR-10 with natural training, and with TRADES adversarial training. These results provide further evidence for the robustness of our optimizer against multiple attack methods, both with natural and adversarial training. We also appreciate the insight of the reviewer regarding the issue with gradient obfuscation, which we intend to explore more thoroughly in our future work.
>
> | Optimizer | AutoAttack | Square |
> | --- | --- | --- |
> | Adam | 20.7 | 23.1 |
> | SNOpt | 16.5 | 19.8 |
> | SGD | 20.4 | 22.8 |
> | c-GTSONO | 26.8 | 29.4 |
>
> | TRADES-λ=0.1 | AutoAttack | Square |
> | --- | --- | --- |
> | SGD | 20.1 | 21.4 |
> | c-GTSONO | 21.8 | 23.9 |
>
> ---
> ---
>
> ### References
> 1. Sun, W., Pan, Y., Lim, J., Theodorou, E. A., & Tsiotras, P. (2018). Min-max differential dynamic programming: Continuous and discrete time formulations. Journal of Guidance, Control, and Dynamics, 41(12), 2568-2580.

---

> ### Author Response · Authors · 2023-11-21
> **Follow up to Reviewer L7bT**
>
> As we are approaching the end of author-reviewer discussion, we would appreciate the reviewer to clarify if our responses has sufficiently addressed the raised concerns, and, if so, we kindly appreciate the reviewer to re-consider their rating.

---

> > ### Comment · Reviewer_L7bT · 2023-11-23
> >
> > Thank you for the detailed reply. For weakness 1, I think it'd be good to write down the formulation in [1], where exactly are the uncertainties and disturbances added, on the system parameters, the states or the controls, and mapping the formulation with input adversarial robustness. As the authors mentioned, for input perturbations, it may be sufficient to only include antagonizing set of controls in the convolutional layers rather than including them for all the weights. I think decomposing the source of disturbances will also provide some insights on where we need the antagonizing set of controls.

---

### Official Review · Reviewer_veDw · 2023-11-05

**Soundness:** 3 good
**Presentation:** 3 good
**Contribution:** 2 fair
**Rating:** 6
**Confidence:** 4

**Summary:**

This paper studies robust optimisation method for neural ODEs. The authors interpret Neural ODE optimization as a min-max optimal control problem, and then design a Game Theoretic Second-Order Neural Optimizer (GTSONO), based on min-max Differential Dynamic Programming, with convergence guarantees to local saddle points. The authors also conduct experiments to verify the performance of GTSONO.

**Strengths:**

This paper is well motivated - addressing the vulnerability to adversarial attacks in neural network related methods, including neural ODE.

The authors design a Game Theoretic Second-Order Neural Optimizer as a robust optimiser for neural ODEs.

They also provide rigorous, theoretical analysis for the proposed method. The proofs are given in detail.

The paper is well written.

**Weaknesses:**

I am worried about the novelty, after reading the calculations. Leveraging min-max methods for adversarial learning is a usual approach.  convergence. The calculations of gradients and backpropagation are simple calculus and linear algebra. The proof of convergence is a direct application of existing results in optimisation. I suggest the authors clarify the novelty of their algorithms and proofs, and discuss the differences and advantages of their method.

The experiments are only conducted on CIFAR-10 and SVHN. Experiments on CIFAR-100, and ImageNet (or at least TinyImageNet) are needed for comparison.

From Tables 1 and 2, GTSONO has fairly bad performance in CIFAR-10 in term of natural accuracy. Please discuss why this happens.

The authors only compare with SGD, Adam, and a second order baseline SNOpt. It is not enough. Please compare your method with most existing methods. I list some below:

https://proceedings.mlr.press/v162/rodriguez22a/rodriguez22a.pdf

https://arxiv.org/pdf/2210.16940.pdf

**Questions:**

Please address the above.

---

> ### Author Response · Authors · 2023-11-18
> **Author Response to Reviewer veDw (part 1/2)**
>
> We would like to thank the reviewer for their positive comments, as well as their constructive criticism and further suggestions for improving our work. In the following we address the points raised by the reviewer.
>
> ### **1. Novelty of Proposed Approach**
> - We first emphasize that our approach differs considerably from conventional min-max methods for adversarial learning. While most min-max adversarial training methods consider the adversary input, our proposed method instead consider the disturbance being injected through the auxiliary **weights.** This, as also recognized by Reviewer gukw, offers a novel perspective by recasting the Neural ODE optimization as a robust trajectory optimization through the lens of min-max optimal control theory. This key reformulation enables computationally efficient and scalable trajectory optimizers (such as min-max Differential Dynamic Programming presented in our work) that is otherwise absent in prior adversarial formulations. We urge the reviewer to recognize the distinction.
> - The experiments empirically verify our choice as our algorithm converges to a saddle point even when trained with clean images, which is translated to an increased robustness of our model against adversarial attacks, even with natural training.
> ---
> ### **2. Convergence proof**
>
> - While we do acknowledge that our convergence analysis is partly inherited from prior theoretical optimization tools, we emphasize that it holds merit to show that our optimizer has theoretical convergence guarantees to a local saddle point. This kind of (local) convergence analysis is crucial in characterizing training stability, especially in deep learning-based algorithms, yet are absent in those inspired by optimal-control methodologies. Furthermore, we highlight that the convergence of our algorithm is still not a direct application of the known convergence results as we need to leverage the formulation of our update law, and the preconditioning with the inverse of the Jacobian of G, for proving that the spectral norm of the Jacobian of our update law is guaranteed to be less than 1 - which implies stable convergence to the saddle point.
> ---
> ### **3. Experiments on CIFAR-100 and Tiny ImageNet and comparison with other methods**
>
> - Following the suggestions of the reviewer, we present further comparisons against the suggested optimizers  Lyanet[1], and FI-ODE[2], as well as on the additional datasets: CIFAR-100, and TinyImageNet. We observe that GTSONO outperforms all baselines , in every tested dataset, while taking slightly longer time to train than SGD. Furthermore, we observe that the next best in performance are LyaNet and FI-ODE, which however have significantly more parameters, leading to dramatically longer training time. Finally, we would like to emphasize that the experiments on the optimizers LyaNet, and FI-ODE are conducted with the architectures and hyperparmeter configurations selected by [1], and [2].
>
> | CIFAR10-Optimizer | PGD_20^0.03 | PGD_20^0.05 | Time (min:sec) | # of Parameters |
> | --- | --- | --- | --- | --- |
> | LyaNet* | $46.1\pm0.4$ | $32.7\pm0.6$ | 32:44 | 19.6 M |
> | FI-ODE* | $48.5\pm0.3$ | $33.4\pm0.6$ | 143:17 | 2.4 M |
> | c-GTSONO | $50.6 ± 0.3$| $35.0 \pm 0.2$ | 3:53 | 1.35M |
>
> | Optimizer | CIFAR100-PGD_20^0.03 | Tiny-ImageNet PGD_20^0.03 | Training TinyImageNet (min:sec) | # of Parameters |
> | --- | --- | --- | --- | --- |
> | Adam | $10.7\pm1.1$ | $1.9\pm0.1$ | 5:37 | 1.35 M |
> | SNOpt | $19.7\pm0.6$ | $3.6\pm0.9$ | 5:41 | 1.35 M |
> | SGD | $9.6\pm0.5$ | $0.8\pm0.2$ | 5:20 | 1.35 M |
> | LyaNet* | $20.1\pm0.5$ | $5.4\pm0.4$ | 313:21 | 19.6 M |
> | FI-ODE* | $15.4\pm0.4  $ | $4.0\pm0.2$ | 607:56 | 2.4 M |
> | c-GTSONO | $23.5\pm0.1$ | $9.1\pm0.8$ | 6:35 | 1.47M |
> *: Different architecture

---

> ### Author Response · Authors · 2023-11-18
> **Author Response to Reviewer veDw (part 2/2)**
>
> ### **4. GTSONO natural accuracy in CIFAR-10**
>
> -  We acknowledge that indeed GTSONO demonstrates a decrease in natural accuracy towards achieving high robustness. Note that this is a well-known trade-off between robustness and accuracy, as studied in [3].  Therefore since our focal focal point is the increased robustness of our optimizer it follows that a decrease in the natural accuracy of our optimizer may be observed. Another interpretation of this phenomenon is that the value function of GTSONO converges to a saddle point, due to the effect of the antagonizing control . This can be viewed as a compromise in the natural accuracy  of the optimizer when compared to other optimizers which converge to a local minimum. It should be noted that the value functions of GTSONO and the baseline optimizers are not the same, therefore this can not serve for direct comparisons, however it provides us with an intuition for this phenomenon.
>
> ---
>
> ### References
> 1. Rodriguez, I. D. J., Ames, A., & Yue, Y. (2022, June). LyaNet: A Lyapunov framework for training neural ODEs. In International Conference on Machine Learning (pp. 18687-18703). PMLR.
> 2. Huang, Y., Rodriguez, I. D. J., Zhang, H., Shi, Y., & Yue, Y. (2022). Fi-ode: Certified and robust forward invariance in neural odes. arXiv preprint arXiv:2210.16940.
> 3. Zhang, H., Yu, Y., Jiao, J., Xing, E., El Ghaoui, L., & Jordan, M. (2019, May). Theoretically principled trade-off between robustness and accuracy. In International conference on machine learning (pp. 7472-7482). PMLR.

---

> ### Author Response · Authors · 2023-11-21
> **Follow up to Reviewer veDw**
>
> As we are approaching the end of author-reviewer discussion, we would appreciate the reviewer to clarify if our responses has sufficiently addressed the raised concerns, and, if so, we kindly appreciate the reviewer to re-consider their rating.

---

> > ### Comment · Reviewer_veDw · 2023-11-22
> >
> > Thanks for your response.
> >
> > I really cannot find "the disturbance being injected through the auxiliary weights" in either your paper or Reviewer gukw's comment. I recognise the novelty of modelling neural ODE by mini-max optimal control, but some papers have modelling neural ODE by optimal control (see https://arxiv.org/pdf/2210.11245.pdf, https://arxiv.org/pdf/1912.05475.pdf).
> >
> > I appreciate the new experiments.
> >
> > My concern on theory remains.
> >
> > Overall, I agree to raise the score to 6.

---

> > > ### Author Response · Authors · 2023-11-22
> > > **Author Response to Reviewer veDw**
> > >
> > > We are glad that the reviewer acknowledged our new experiment results, and greatly appreciate the decision to increase the rating. We provide additional clarifications below.
> > >
> > > - **Disturbance being injected through the auxiliary weights**
> > >
> > >      We kindly refer the reviewer to Sec 3.1 where we introduce the problem formulation and how the antagonizing set of controls, inspired by game theoretic optimal control, enables us to represent the disturbances in the system. This prompted us to expand the already established connection between Neural ODEs and continuous optimal control by establishing connections between adversarial defense methods and robust optimal control.
> > >
> > > - **Other works modeling Neural ODE by optimal control**
> > >
> > >     We first thank the reviewer for the additional references. We do acknowledge the existence of numerous prior works, including those already cited in our Sec 1, on modeling neural ODE by optimal control. In fact, it is this very connection that motivates our work, whose primary contribution is to *extend* the optimal control framework to robust trajectory optimization using minimax formulation. This extension results in a *new* computational framework that is particularly well-suited for applications such as adversarial robustness (while the references brought up by the reviewer, albeit interesting, focus on other applications).
> > >
> > > - **Theoretical analysis**
> > >
> > >     We want to emphasize that the primary contribution of our work is the conceptualization of our game theoretic optimizer as described in Sec 3. Subsequently, we proceed to establish through our convergence analysis that our methodology has converge guarantees to a local saddle point that are agnostic of the architecture and hyperparameter configuration. This serves as an additional aid in characterizing the training stability of our optimizer, which is otherwise absent in prior control-theoretic optimizer for training Neural ODEs.

---

### Author Response · Authors · 2023-11-18
**Author response to all reviewers**

We thank the reviewers for their valuable insights. We are excited that the reviewers identified the importance and motivation of the problem (*Reviewer veDw, gukw*), appreciated the novelty of the contributions  (*Reviewer gukw*), acknowledged our theoretical analysis to render second order optimization feasible (Reviewer L7bT), and the convergence guarantee of the proposed optimizer (Reviewers L7bT, gukw) and found the paper well-written (*Reviewer* veDw and gukw).  We believe our method takes a significant step toward principled algorithmic design inspired from optimal control theory to robustify deep continuous-time models.

---

We append a revised version of our manuscript to address concerns raised by Reviewer veDw regarding the novelty of our optimizer. We believe that these comments stem from our contributions not being properly underscored in the Introduction of our paper. Additionally, we rearranged the Expreriments section to present more efficiently the training time and computational resources needed by each optimizer in the main part of the paper following the remarks of Reviewer L7bT.

Furthermore, we carefully addressed each of the comments and concerns raised by the reviewers. Importantly, we provide with clarifications to Reviewer’s gukw concerns about the computational overhead of our optimizer, and the reason it remains computationally more favorable compared to other adversarial training methods. Additionally, we present more experiments, as suggested by each reviewer. More specifically,
- Extensive comparisons between GTSONO and the baseline optimizers on more datasets such as the CIFAR-100 and TinyImageNet, as suggested by Reviewers veDw and gukw.
- Comparisons of GTSONO against state-of-the-art optimizers that enhance robustness, as suggested by Reviewers veDw and gukw.
- Comparisons between GTSONO and baseline optimizers under non-gradient based attacks as suggested by Reviewer L7bT.

We hope that our new presentation has improved the manner in which our contributions in robustifying deep continuous-time models are presented. We try our best to resolve all raised concerns in the individual responses below.

---

### Author Response · Authors · 2023-11-21
**Follow up before the author-reviewer discussion period closes**

In our rebuttal, we gave clarifications to the concerns raised by the reviewers, and in the additional experiments we showed that **GTSONO was the most robust optimizer in every tested dataset**. Specifically, we showed that our optimizer consistently outperformed standard and state-of-the art optimizers that enhance robustness in a variety of datasets, such as CIFAR-10, CIFAR-100, and TinyImageNet. Importantly, it was shown that GTSONO was dramatically more efficient than the other robust optimizers tested.

As we are approaching the end of the discussion period, we would like to ask whether our responses have covered the questions raised in your initial reviews. We are happy to provide any further clarification and discussion. If our replies adequately address your concerns, we would like to kindly ask the reviewers to raise the score, so that it better reflects the discussion at the current stage.

---

### Meta-Review · Area_Chair_KN5C · 2023-12-06

**Metareview:**

This paper studies the robustness of neural ODE. Reviewers appreciate the authors' effort on providing theoretical understanding on this important research question. However, some reviewers still have concerns on the novelty of the theory compared with the prior work after the rebuttal. AC encourages the authors to address this concern in the later version of the paper.

**Justification For Why Not Higher Score:**

AC has tried to lead the discussion. It seems that no reviewer is willing to stand out and champion the paper. Furthermore, Reviewer veDw mentioned his/her concerns on the theory part remains after the rebuttal. The authors fail to address all questions from the reviewers.

**Justification For Why Not Lower Score:**

Though on the borderline, the overall score is positive and no reviewer wants to fight for a rejection.

---

### Decision · Program_Chairs · 2024-01-16

Accept (poster)